# Targeting endothelin 1 receptor-miR-200b/c-ZEB1 circuitry blunts metastatic progression in ovarian cancer

Rosanna Sestito [1], Roberta Cianfrocca[1], Piera Tocci [1], Laura Rosanò[1,2], Andrea Sacconi[3], Giovanni Blandino [3] & Anna Bagnato [1✉]

Identification of regulatory mechanisms underlying the poor prognosis of ovarian cancer is necessary for diagnostic and therapeutic implications. Here we show that endothelin A receptor ($ET_A R$) and ZEB1 expression is upregulated in mesenchymal ovarian cancer and correlates with poor prognosis. Notably, the expression of $ET_A R$ and ZEB1 negatively correlates with miR-200b/c. These miRNAs, besides targeting ZEB1, impair $ET_A R$ expression through the 3'UTR binding. ZEB1, in turn, restores $ET_A R$ levels by transcriptionally repressing miR-200b/c. Activation of $ET_A R$ drives the expression of ZEB1 integrating the miR-200/ZEB1 double negative feedback loop. The $ET_A R$-miR-200b/c-ZEB1 circuit promotes epithelial-mesenchymal transition, cell plasticity, invasiveness and metastasis. Of therapeutic interest, $ET_A R$ blockade with macitentan, a dual $ET_A R$ and $ET_B R$ antagonist, increases miR-200b/c and reduces ZEB1 expression with the concomitant inhibition of metastatic dissemination. Collectively, these findings highlight the reciprocal network that integrates $ET_A R$ and ZEB1 axes with the miR-200b/c regulatory circuit to favour metastatic progression in ovarian cancer.

---

[1] Preclinical Models and New Therapeutic Agents Unit, IRCCS-Regina Elena National Cancer Institute, Rome, Italy. [2] Institute of Molecular Biology and Pathology, National Research Council (CNR), Rome, Italy. [3] Oncogenomic and Epigenetic Unit, IRCCS-Regina Elena National Cancer Institute, Rome, Italy. ✉email: annateresa.bagnato@ifo.gov.it

Ovarian carcinoma is the leading cause of death from gynecologic malignancies[1]. The majority of histological types belong to high-grade serous ovarian carcinoma (HG-SOC), with relatively poor prognosis due to the advanced stage at diagnosis, widespread dissemination into the peritoneal cavity and recurrence[2]. Moreover, the molecular heterogeneity of this disease renders difficult the classification and treatment. Gene expression analyses of HG-SOC patients classified this tumor in four transcriptionally different subtypes: immunoreactive, differentiated, proliferative, and mesenchymal subgroups[3,4]. Among them, the mesenchymal subtype is associated with poor patient survival, remaining the major clinical challenge in ovarian cancer[4,5]. Consequently, the identification of the molecular mechanisms underlying the poor prognosis of HG-SOC patients, especially the mesenchymal subtype, is of paramount importance in order to develop new strategies for the management of this disease.

The epithelial–mesenchymal transition (EMT) is a complex metastasis-related program tightly regulated by the interplay between signaling pathways, microRNA (miRNA), a well-characterized class of small regulatory RNAs, and EMT-transcription factors (EMT-TF)[6–9]. Among these, zinc finger E-box binding homeobox 1 (ZEB1) may directly repress epithelial genes and positively regulate mesenchymal factors[10]. In ovarian cancer, high levels of ZEB1 correlate with loss of E-cadherin[11] and associate with advanced diseases, metastasis and poor prognosis of patients[12–14]. Because of the crucial role of EMT in metastatic progression, the circuit regulating EMT has been studied, including the canonical miR-200/ZEB1 regulatory axis[8]. The miR-200 family, consisting of five members (miR-200a/b/c, miR-141, and miR-429), which form two functional clusters (miR-200a/-141 and miR-200b/c/-429) based upon their seed sequences, is able to downregulate ZEB1 through the direct binding of its mRNA[15]. Forced expression of miR-200 family members represses ZEB1 expression and inhibits the capacity of ovarian cancer cells to undergo migration and invasion, implicating ZEB1 as a miR-200 target[16,17]. Moreover, the EMT-regulatory circuit requires a balanced expression of ZEB1 and miR-200, which are interlinked in a double negative feedback loop[18,19]. In particular, while miR-200 family members cause post-transcriptional repression of ZEB1, the latter inhibits the promoter activity of these miR-200 transcription units[20–22]. A deep understanding of the complexity of the underlying mechanism involved in miR-200 regulatory circuits is necessary to comprehend the dynamic functional relationship endowing cells, with a high degree of cell plasticity necessary for malignant progression. Therefore, the knowledge of the complex ZEB1/miRNA interplay, established by functional links with other signaling pathways[23,24], may provide mechanistic insight into the regulatory networks controlling ovarian cancer aggressiveness and metastatic progression.

It has been widely demonstrated that endothelin-1 (ET-1) axis, including the peptide ligand ET-1 and the two G-protein coupled receptors (ET$_A$R and ET$_B$R), is a potent inducer of EMT and metastatic progression[25,26]. In particular, in ovarian cancer, where high ET$_A$R levels correlate with a poor prognosis, EMT and chemoresistance[27,28], the activation of the ET-1/ET$_A$R pathway controls EMT by regulating the EMT-TF Snail[27]. We have previously underscored a critical miRNA, miR-30a, as a tumor suppressor in ovarian cancer able to target ET$_A$R and to regulate chemoresistance[29]. In a search of the complex regulatory networks driving ovarian cancer metastatization, in this study we identify a miR-200b/c-dependent circuit established between ET$_A$R and ZEB1 that is regulated by ET-1 to induce invasive cell behavior fostering metastatic progression. Therefore, the interruption of this circuit can be

exploited to develop efficient strategies for preventing metastasis and recurrence, leading to an improvement of the survival of ovarian cancer patients.

## Results

### The expression of ET$_A$R/ZEB1 is upregulated in ovarian cancers and correlates with poor prognosis.
Having demonstrated that ET-1/ET$_A$R axis is a critical driver of EMT in ovarian cancer[26,27], we sought to investigate whether ET-1/ET$_A$R axis could be associated with the canonical EMT-TF ZEB1. The analysis of ET$_A$R and ZEB1 protein expression in a panel of ovarian cancer cells revealed their positive correlation (Fig. 1a, b). In particular, ET$_A$R and ZEB1 resulted overexpressed in mesenchymal cell lines (HEY, SKOV3), with higher expression of vimentin and N-cadherin mesenchymal markers, than epithelial cell lines (CAOV3, OVCA 433) (Fig. 1a). To confirm the association between ET$_A$R and ZEB1, we performed an analysis from The Cancer Genome Atlas (TCGA) database of 535 miRNA/mRNA matched high-grade serous ovarian cancer (HG-SOC) patients, which are subdivided in four subtypes (proliferative, mesenchymal, differentiated and immunoreactive) based on their specific gene expression profiles[4]. Importantly, higher mRNA levels of ET$_A$R and ZEB1 were observed in the mesenchymal subtype of patients compared to the other subgroups ($p = 2.0441e-48$ and $p = 6.64e-40$, respectively) (Fig. 1c). Moreover, the expression of ET$_A$R and ZEB1 was positively correlated in the mesenchymal subtype ($n = 106$, R Pearson = 0.42, $p = 6.48e-06$) as well as in the entire cohort of HG-SOC patients ($n = 535$, R Pearson = 0.54, $p = 6.33e-42$). To assess the prognostic relevance of the ET$_A$R and ZEB1 integration, TCGA cohort was dichotomized into patients expressing combined high *versus* low expression of these two genes. Importantly, the coexpression of *EDNRA* (ET$_A$R) and *ZEB1* genes was associated with poor prognosis of HG-SOC patients, in terms of overall survival (OS; hazard ratio (HR) 1.36 [CI 95%: 1.03–1.78], $p = 0.029$) (Fig. 1d and Supplementary Fig. 1a) and progression-free survival (PFS; HR 1.52 [CI 95%: 1.16–1.99], $p = 0.002$) (Fig. 1e and Supplementary Fig. 1b). Altogether, these results highlight a potential interplay between ET$_A$R and ZEB1 in the control of clinical outcome of ovarian cancer.

### ET$_A$R and ZEB1 are direct transcriptional targets of miR-200b/c.
The key miRNA canonically involved in the EMT-regulatory circuits, such as members of the miR-200 family, are downregulated during EMT[8]. To identify potential miRNAs, which concomitantly target ET$_A$R and ZEB1, we examined the overlap between miRNA predicted by miRanda[30], TargetScan[31], and miRmap[32] bioinformatic tools. Interestingly, this analysis revealed that the 3′UTR of ET$_A$R and ZEB1 contained complementary matching regions for miR-200b/c and miR-429, belonging to the miR-200 family members, along with miR-200a and miR-141 (Supplementary Fig. 2a). As expected, all the analyzed mesenchymal-like cells expressed low levels of these miRNAs (Fig. 2b and Supplementary Fig. 2b). As predicted by bioinformatic analysis, we observed that miR-141 and miR-200a, that were able to target ZEB1, had no detectable effect on ET$_A$R protein expression (Supplementary Fig. 2c). To deepen further the analysis, we examined a 35-miRNA-based signature (MiR-OvaR), as a promising predictor model of risk of ovarian cancer relapse or progression[33]. Among these 35 miRNAs, 16 miRNAs, whose downregulation associates with a worse prognosis of ovarian cancer patients[33], have been analyzed revealing that miR-200b/c and miR-429 could be putative regulatory miRNAs of both ET$_A$R and ZEB1, as depicted in the Venn diagram (Fig. 2a). In light of this clinical relevant analysis, we focused our study on

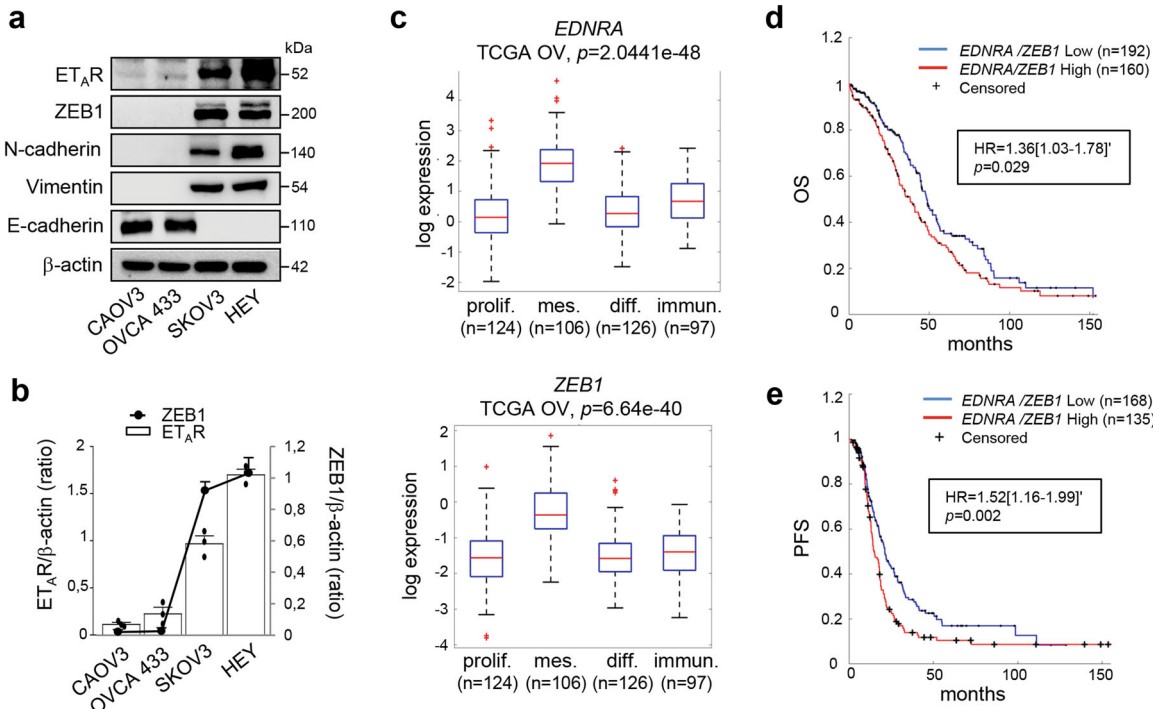

**Fig. 1 The expression of ET$_A$R/ZEB1 is upregulated in ovarian cancers and correlates with poor prognosis. a** Expression of ET$_A$R, ZEB1, N-cadherin, Vimentin, and E-cadherin in ovarian cancer cell lines analyzed by Western blotting (WB). β-actin is used as loading control. **b** Correlation of the ET$_A$R and ZEB1 expression in the indicated cell lines. The ratio of ET$_A$R/β-actin and ZEB1/β-actin, evaluated by WB as shown in **a**, is presented as bar and line, respectively. Values are the mean ± SD ($n = 3$). **c** Boxplot diagrams of *EDNRA* (ET$_A$R) and *ZEB1* gene expression in HG-SOC patients from TCGA, subdivided in four molecular subtypes. **d** Kaplan–Meier curves of overall survival (OS) of 352 patients from TCGA grouped in high *EDNRA/ZEB1* levels (160 patients, $z$-score > 1.5, red line) and low *EDNRA/ZEB1* expression levels (192 patients, $z$-score < 1.5, blue line). **e** Kaplan–Meier curves of progression-free survival (PFS) of 303 patients from TCGA subdivided in high levels of *EDNRA/ZEB1* expression (135 patients, $z$-score > 1.5, red line) and low levels of *EDNRA/ZEB1* (168 patients, $z$-score < 1.5, blue line).

miR-200b/c and profiled their expression in the panel of ovarian cancer cells. In agreement with the computational results, a negative association between miR-200b/c and ET$_A$R expression has been observed (Fig. 2b). Accordingly, a negative correlation between ET$_A$R/miR-200b and ET$_A$R/miR-200c coexpression has been found in the HG-SOC patients from TCGA ($n = 535$, R Pearson = -0.25, $p = 5.99e{-}09$ and $n = 535$, R Pearson = -0.2, $p = 3.87e{-}06$, respectively). To further demonstrate the inhibitory role of these miRNAs on ET$_A$R, we found that miR-200b/c ectopic expression, using specific mimic-miRNAs, significantly reduced ET$_A$R expression at both mRNA (Fig. 2c) and protein (Fig. 2d and Supplementary Fig. 2d) levels. Interestingly, also miR-429 was able to reduce both ZEB1 and ET$_A$R expression (Supplementary Figure 2c). To determine whether ET$_A$R is a direct target of miR-200b/c, we performed luciferase assays using the wild type ET$_A$R 3′UTR reporter plasmid and its mutated derivative on the miR-200b/c seed sequences (Supplementary Fig. 2e, f). Unlike for the mutated ET$_A$R 3′UTR reporter plasmid, ectopic miR-200b/c significantly reduced the ET$_A$R reporter activity thereby indicating the potential of the ET$_A$R 3′UTR to be targeted by miR-200b/c (Fig. 2e and Supplementary Fig. 2g). Moreover, miR-200b/c silencing, using specific miRNA inhibitors (anti-miR-200b/c), increased ET$_A$R expression, as well as cell viability (Fig. 2f, g). On the contrary, miR-200b/c overexpression decreased cell viability similarly to the effect elicited by the ET$_A$R silencing or by the pharmacological block with macitentan, a potent ET$_A$R antagonist with significant affinity for ET$_B$R[25] (Fig. 2g). Of note, in cells transfected with a plasmid encoding ET$_A$R lacking the 3′UTR, miR-200b/c lost their capability to reduce both ET$_A$R expression and cell proliferation compared to

cells transfected with wt ET$_A$R (Fig. 2h, i), indicating a negative control of miR-200b/c over ET$_A$R expression.

As miR-200 regulation is critical for the key EMT-TF ZEB1 expression, we next examined the underlying mechanism driving ZEB1 expression in ovarian cancer. A negative association between ZEB1/miR-200b and ZEB1/miR-200c expression has been observed in the analyzed cells (Fig. 3a), as well as in HG-SOC patients from TCGA dataset ($n = 535$, R Pearson = -0.28, $p = 2.48e{-}11$ and $n = 535$, R Pearson = -0.24, $p = 2.44e{-}08$, respectively). Ectopic miR-200b/c strongly reduced ZEB1 mRNA (Fig. 3b) and protein (Fig. 3c and Supplementary Fig. 2d) expression levels. In line with the evidence that miR-200 family members regulate ZEB1 at post-transcriptional level through the direct binding of the ZEB1 3′UTR[24], ectopic miR-200b/c, whose seed sequences are present in the ZEB1 3′UTR (Supplementary Fig. 2e), reduced the luciferase activity of a ZEB1 3′UTR reporter plasmid (Fig. 3d), while silencing of miR-200b/c enhanced the expression of ZEB1 (Fig. 3e). Collectively, these findings demonstrate that miR-200b/c are involved in the regulation of ET$_A$R and ZEB1 expression in ovarian cancer cells.

**ZEB1 regulates ET$_A$R expression through the suppression of miR-200b/c.** Besides to the miR-200b/c-dependent regulation of ZEB1, it is known that ZEB1, in turn, is able to suppress the transcription of miR-200 family members, establishing a reciprocal ZEB1/miR-200 feedback loop that functions as a molecular switch to control cell plasticity[18–23]. Accordingly, an increase of both miR-200b/c levels was detected in ZEB1-silenced cells (Fig. 4a and Supplementary Fig. 3a). Because forced expression of miR-200

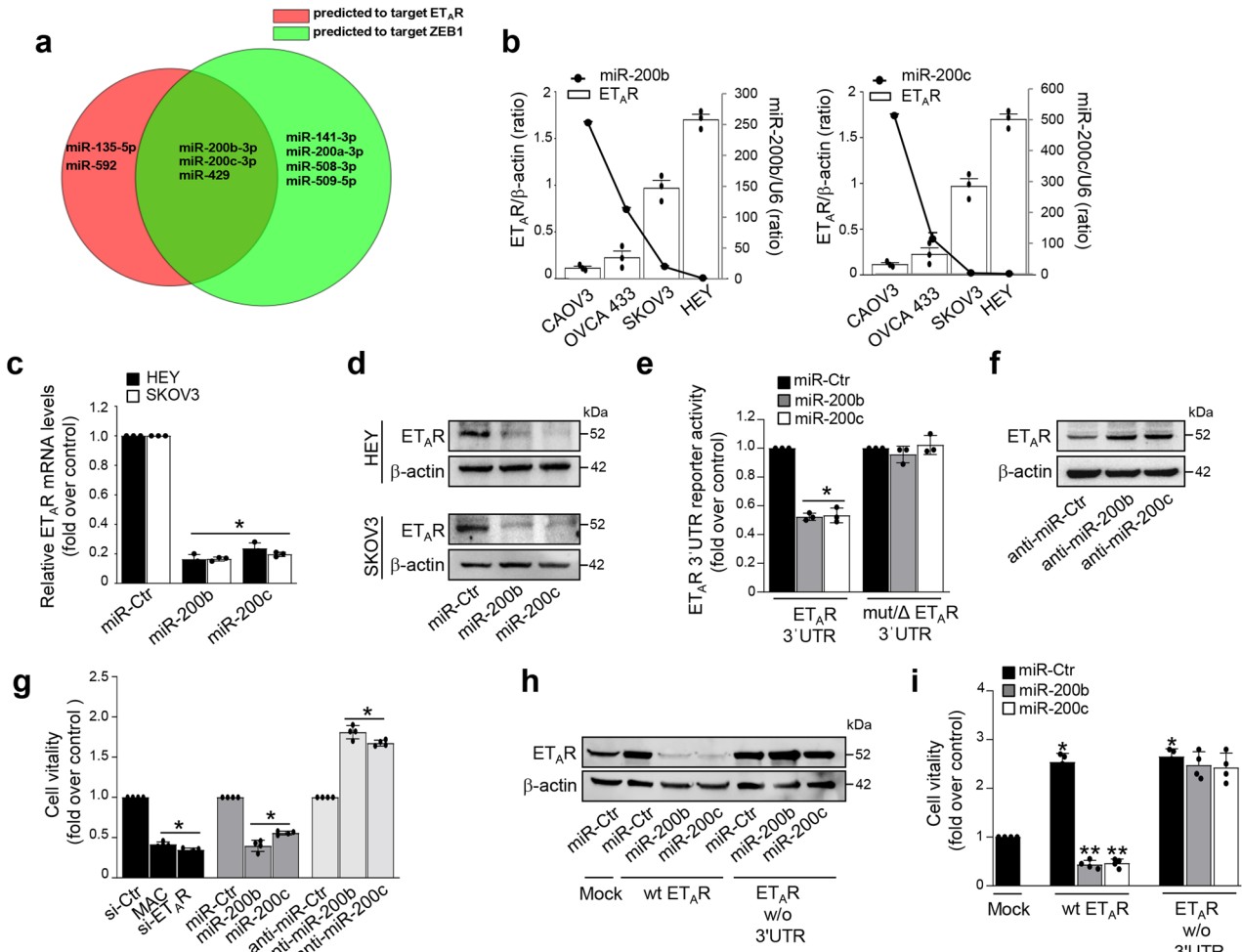

**Fig. 2 ET_AR is a direct target of miR-200b/c. a** Venn diagram of miRNAs whose expression is associated with a good prognosis in ovarian cancer patients (MiROvaR signature[33]) showing the overlap ($n = 3$) between miRNA predicted to target the ET_AR 3′UTR ($n = 5$, red circle) and miRNA predicted to target the ZEB1 3′UTR ($n = 7$, green circle). **b** Expression levels of ET_AR and miR-200b or miR-200c in the indicated cell lines. The ratio of ET_AR/β-actin, evaluated by WB and miR-200b/c expression, evaluated by qRT-PCR and normalized to U6, are shown as bar and line, respectively. Values are the mean ± SD ($n = 3$). **c** ET_AR mRNA levels in HEY and SKOV3 cells transfected for 48 h with mimic-miRNA control (miR-Ctr), mimic-miR-200b (miR-200b) or mimic-miR-200c (miR-200c) evaluated by qRT-PCR and normalized to cyclophilin-A. Values are the mean ± SD ($n = 3$; *$p < 0.001$ vs. Ctr). **d** Lysates from HEY and SKOV3 cells transfected as in **c** are analyzed by WB for ET_AR expression. β-actin is used as loading control. **e** Luciferase activity in HEY cells co-transfected for 48 h with miR-Ctr, miR-200b, or miR-200c together with the reporter plasmid containing the 3′UTR region of ET_AR (ET_AR 3′UTR) or its triple mutant (mut/ΔET_AR 3′UTR). Values are the mean ± SD expressed as fold induction ($n = 3$; *$p < 0.001$ vs. Ctr). **f** WB for ET_AR expression in the lysates from HEY cells 48 h transfected with miRNA control, miR-200b or miR-200c inhibitors (anti-miR-Ctr, anti-miR-200b, and anti-miR-200c). β-actin is used as loading control. **g** Cell viability of HEY cells, transfected as in **c** and **f**, or with siRNA control (si-Ctr), or si-ET_AR, and treated or not with macitentan (MAC) for 48 h. Values are the mean ± SD ($n = 4$; *$p < 0.001$ vs the respective Ctr). **h** Lysates of HEY cells transfected as in **c** together with a plasmid control (Mock), a wt ET_AR expression plasmid, or with an ET_AR expression plasmid w/o the 3′UTR as indicated are analyzed by WB for ET_AR expression. β-actin is used as loading control. **i** Cell vitality of HEY cells transfected as in **h**. Values are the mean ± SD expressed as fold induction ($n = 4$; *$p < 0.001$ vs. Mock; **$p < 0.001$ vs. wt ET_AR).

reduced ET_AR levels, we then evaluated whether ZEB1 could also affect ET_AR expression. ZEB1 depletion significantly decreased the expression of mRNA and protein levels of ET_AR (Fig. 4b and Supplementary Fig. 3b, c). Similarly, the abrogation of ZEB1 transcriptional activity, by using a Flag-tagged construct able to recognize and block the DNA binding domain of ZEB1 (DB-ZEB1-Flag)[34], upregulated miR-200b/c and, in parallel, reduced ET_AR expression (Supplementary Fig. 3d, e). These results suggest that ZEB1 could control ET_AR expression through the miR-200b/c inhibition. To validate this hypothesis, we analyzed the effect of ZEB1 manipulation on the ET_AR 3′UTR reporter activity. ZEB1 overexpression downregulated miR-200b/c and, in parallel, increased the ET_AR 3′UTR reporter activity (Fig. 4c, d). In line with these results, forced expression of miR-200b/c reduced the activity

of the ET_AR reporter gene and abrogated the stimulatory effect of ZEB1 (Fig. 4c, d). Similar findings were obtained by the analysis of ET_AR protein expression (Fig. 4e). On the contrary, in ZEB1-depleted cells, whose levels of miR-200b/c were increased, we observed a reduction of the ET_AR 3′UTR reporter activity (Fig. 4f, g). In these cells, the silencing of miR-200b/c, by using anti-miR-200, enhanced the activity of the ET_AR 3′UTR reporter, that was not perturbed by the concomitant silencing of ZEB1 (Fig. 4f, g). Importantly, ZEB1 depletion was unable to decrease the activity of the ET_AR 3′UTR reporter lacking functional miR-200b/c binding sites (Fig. 4h), demonstrating that these miRNAs play a role in the ZEB1-mediated regulation of ET_AR. Altogether, these findings reveal ET_AR as an integrated component of the miR-200/ZEB1 reciprocal feedback loop (Fig. 4i).

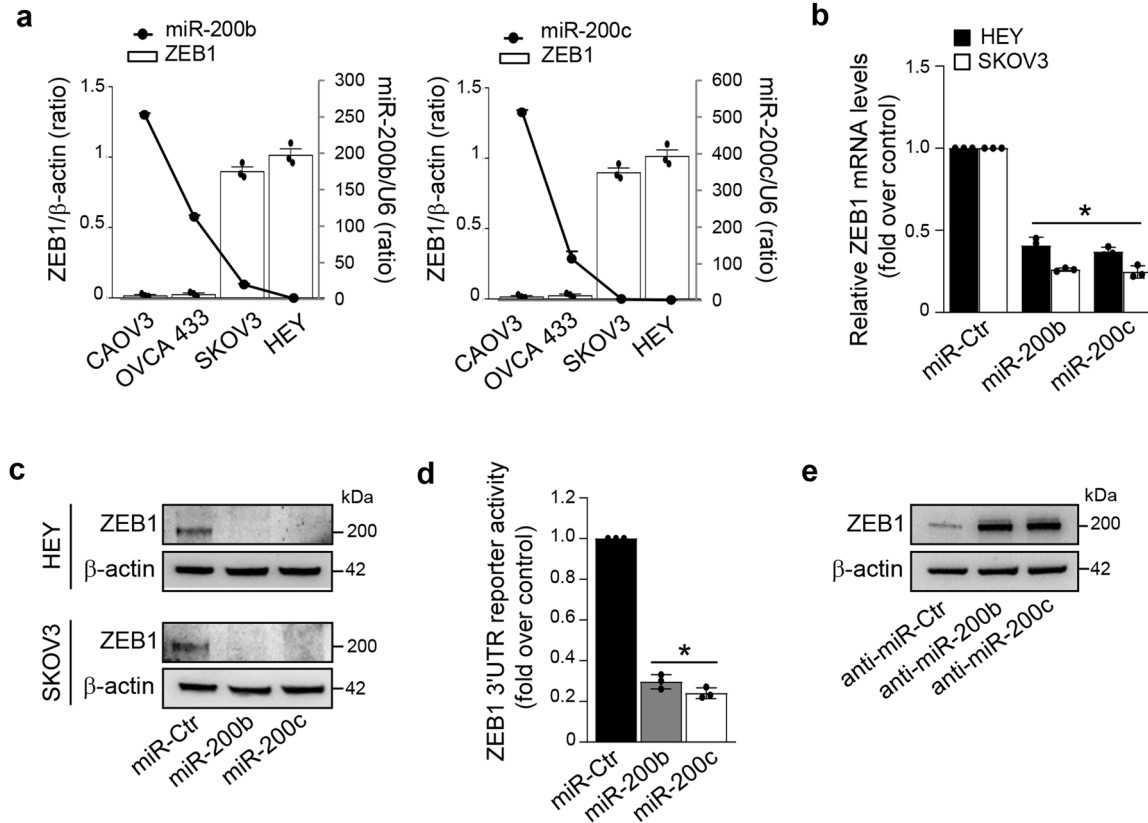

**Fig. 3 miR-200b/c downregulate ZEB1. a** Expression levels of ZEB1 and miR-200b or miR-200c in the indicated cell lines. The ratio of ZEB1/β-actin, evaluated by WB and miR-200b/c expression, evaluated by qRT-PCR, are shown as bar and line, respectively. Values are the mean ± SD ($n = 3$). **b** ZEB1 mRNA levels in HEY and SKOV3 cells transfected for 48 h with mimic-miRNA control (miR-Ctr), mimic-miR-200b (miR-200b), or mimic-miR-200c (miR-200c) evaluated by qRT-PCR and normalized to cyclophilin-A. Values are the mean ± SD ($n = 3$; $*p < 0.001$ vs. Ctr). **c** Lysates from HEY and SKOV3 cells transfected as in **b** are analyzed by WB for ZEB1 expression. β-actin is used as loading control. **d** Luciferase activity in HEY cells co-transfected for 48 h with miR-Ctr, miR-200b, or miR-200c together with the reporter plasmid containing the 3′UTR region of ZEB1. Values are the mean ± SD ($n = 3$; $*p < 0.001$ vs. Ctr). **e** WB for ZEB1 expression in the lysates from HEY cells transfected for 48 h with miRNA control, miR-200b, or miR-200c inhibitors (anti-miR-Ctr, anti-miR-200b, and anti-miR-200c). β-actin is used as loading control.

**ET-1/ET$_A$R axis downregulates miR-200b/c via ZEB1.** In light of the transcriptional regulator activity of ET-1 on mediators of EMT[26,27], we next evaluated whether ET-1/ET$_A$R axis could modulate the expression of ZEB1. ET-1 stimulation in mesenchymal cells upregulated ZEB1 expression at both mRNA (Supplementary Fig. 4a) and protein levels (Fig. 5a and Supplementary Fig. 4b, c), as well as ZEB1 promoter activity (Fig. 5b and Supplementary Fig. 4e), indicating that ET-1/ET$_A$R axis positively regulates ZEB1 expression at the transcriptional level. Consistently, either in cells treated with macitentan, or in ET$_A$R-silenced cells, ET-1 was unable to modulate the expression of ZEB1 or its promoter activity (Fig. 5a, b and Supplementary Fig. 4c, e). Similarly, treatment with the selective ET$_A$R antagonist zibotentan (ZD4054), but not with the selective ET$_B$R antagonist BQ788, inhibited the ET-1-driven induction of ZEB1 protein levels, clearly demonstrating the specific role of ET$_A$R in the regulation of ZEB1 expression (Supplementary Fig. 4d). Next, we examined whether the ET-1-mediated ZEB1 regulation could also impact on miR-200b/c expression. Importantly, in ovarian cancer cells stimulated with ET-1 a reduction of these miRNAs was observed (Fig. 5c and Supplementary Fig. 4f, g). ET$_A$R blockade by macitentan impaired the ET-1 ability to reduce miR-200b/c levels (Fig. 5c and Supplementary Fig. 4g). In the absence of ZEB1, the expression of miR-200b/c was increased in both ET-1-stimulated and unstimulated cells (Fig. 5c and Supplementary Fig. 4g). Accordingly, similar results were obtained by using the

DB-ZEB1-Flag plasmid (Supplementary Fig. 4h, i), indicating that ET-1/ET$_A$R axis controls the expression of these miRNAs through the reciprocal relationship with ZEB1. The biogenesis of miRNAs is a multistep process that begins with the miRNA gene transcription by the RNA Polymerase II and the production of primary miRNA transcripts, named pri-miRNAs[35]. Therefore, to evaluate whether ET-1 may regulate miR-200b/c transcription, we then analyzed their pri-miRNAs and, intriguingly, observed a significant reduction of both pri-miR-200b/c in ET-1-stimulated cells (Fig. 5d). This ET-1-induced effect was lost in cells treated with macitentan and in ZEB1-silenced cells (Fig. 5d), revealing a ZEB1-dependence on the ET-1-mediated pri-miR-200b/c regulation. To further assess the ET-1-mediated effect on miR-200b/c transcription, the reporter plasmids, containing sequences of the miR-200b[21] and miR-200c[22] promoters and including ZEB1 binding sites, were used (Fig. 5e). ET-1 significantly reduced the activity of both miR-200 promoters that was recovered by macitentan, as well as by ET$_A$R silencing (Fig. 5f and Supplementary Fig. 4j). Importantly, in ZEB1 silenced cells, miR-200b/c transcriptional activity increased and the addiction of ET-1 did not influence the promoter activity of these miRNAs (Fig. 5f and Supplementary Fig. 4j), indicating that ZEB1 is required to mediate ET-1-induced effects. We next sought to determine whether ET-1 might favor the physical interaction of ZEB1 with the miR-200 promoters. We performed chromatin immunoprecipitations (ChIP) on SKOV3 cells using antibodies against ZEB1

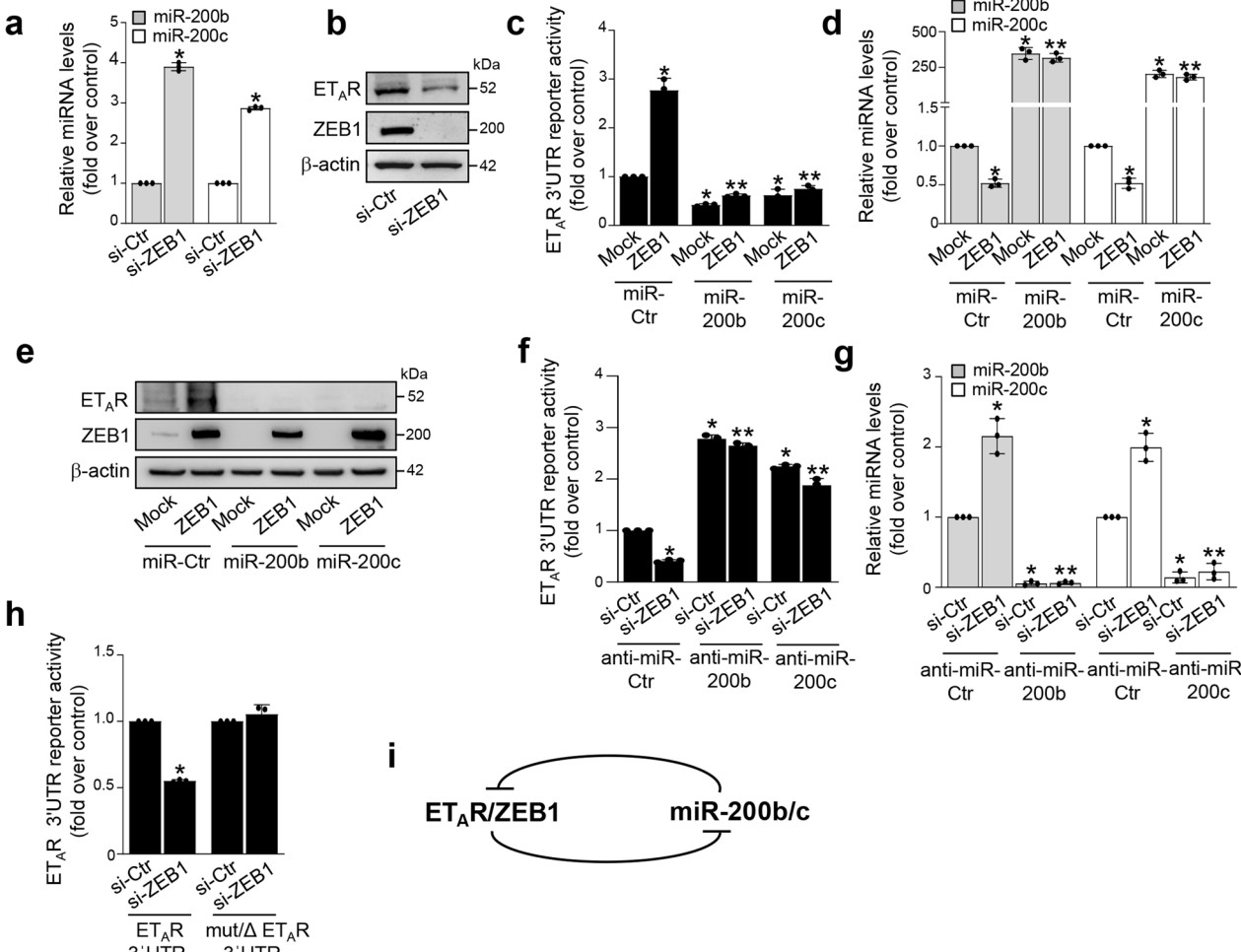

**Fig. 4 ZEB1 regulates ET$_A$R expression through the repression of miR-200b/c. a, b** Expression of miR-200b/c and ET$_A$R, ZEB1 proteins in HEY cells transfected for 72 h with siRNA control (si-Ctr) or si-ZEB1 as determined by qRT-PCR (**a**) and WB (**b**). U6 and β-actin are used to normalize miRNA and protein expression, respectively. Values are the mean ± SD ($n = 3$; *$p < 0.0001$ vs. Ctr). **c, d** Luciferase activity (**c**) in HEY cells co-transfected for 48 h with the indicated mimic-miRNAs, in the presence or in the absence of ZEB1 plasmid together with the ET$_A$R 3′UTR reporter plasmid. qRT-PCR (**d**) of the miR-200b/c expression in cells transfected as above and normalized to U6. Values are the mean ± SD expressed as fold induction ($n = 3$; *$p < 0.01$ vs. Ctr; **$p < 0.001$ vs. ZEB1-transfected cells). **e** Expression of ET$_A$R and ZEB1 proteins in lysates from cells treated as in **c** analyzed by WB. β-actin is used as loading control. **f, g** Luciferase activity (**f**) in HEY cells co-transfected for 48 h with the indicated anti-miRNAs, in the presence or absence of si-ZEB1 together with the ET$_A$R 3′UTR reporter plasmid. qRT-PCR (**g**) of the miR-200b/c expression, in cells transfected as above and normalized to U6. Values are the mean ± SD expressed as fold induction ($n = 3$; *$p < 0.01$ vs. Ctr; **$p < 0.001$ vs. si-ZEB1-transfected cells). **h** Luciferase activity in HEY cells co-transfected for 48 h with si-Ctr or si-ZEB1 together with the reporter plasmid containing the 3′UTR region of ET$_A$R (ET$_A$R 3′UTR) or its triple mutant (mut/ΔET$_A$R 3′UTR). Values are the mean ± SD expressed as fold induction ($n = 3$; *, $p < 0.001$ vs. ET$_A$R 3′UTR-transfected Ctr cells). **i** Schematic model of the ET$_A$R/ZEB1 and miR-200b/c feedback circuit.

and IgG as a control for nonspecific binding. In cells stimulated with ET-1, we found a strong enrichment of ZEB1 on the miR-200b promoter, indicating the capability of ET-1 to mediate ZEB1 regulation on the transcriptional activity of miR-200b. As a further negative control for nonspecific enrichment, we assayed the immunoprecipitates for a region −1900 bp upstream the miR-200b TSS site (lacking ZEB1 binding sites). No enrichment was found at this control region, indicating the specific recruitment of ZEB1 on the promoter of the miR-200b locus (Supplementary Fig. 4k). Altogether, these findings demonstrate that activation of ET$_A$R by ET-1 decreases miR-200b/c levels by suppressing their transcription via ZEB1 in a complex and interdependent way.

**The integrated ET$_A$R-miR-200b/c-ZEB1 circuit is involved in ET-1-mediated ovarian cancer aggressiveness.** Next, we sought to investigate the aggressive features induced by the integrated

ET-1/ET$_A$R and ZEB1/miR-200 axes. Since enhanced activity of matrix metalloproteases (MMPs) is a key feature of the ET-1-induced invasive phenotype of ovarian cancer cells[36], we performed gelatin zymography analyses to evaluate the ZEB1 capability to activate MMPs. In cells unstimulated and stimulated with ET-1, the silencing of ZEB1 reduced MMP-2/9 activation, likewise the treatment with macitentan. Interestingly, forced expression of miR-200b/c in cells inhibited the activation of MMP-2/9 in basal, as well as upon ET-1-stimulation (Fig. 6a).

In line with the above results, the ET-1-induced expression of mesenchymal markers, N-cadherin and vimentin, and reduced expression of E-cadherin were inhibited upon ZEB1 depletion, or ectopic expression of miR-200b/c, or following macitentan treatment, at mRNA and protein levels (Fig. 6b, c and Supplementary Fig. 5a). Similarly, miR-200b/c overexpressions, as well as ET-1R blockade, were also able to inhibit the ET-1-dependent ZEB1 upregulation (Fig. 6b, c and Supplementary

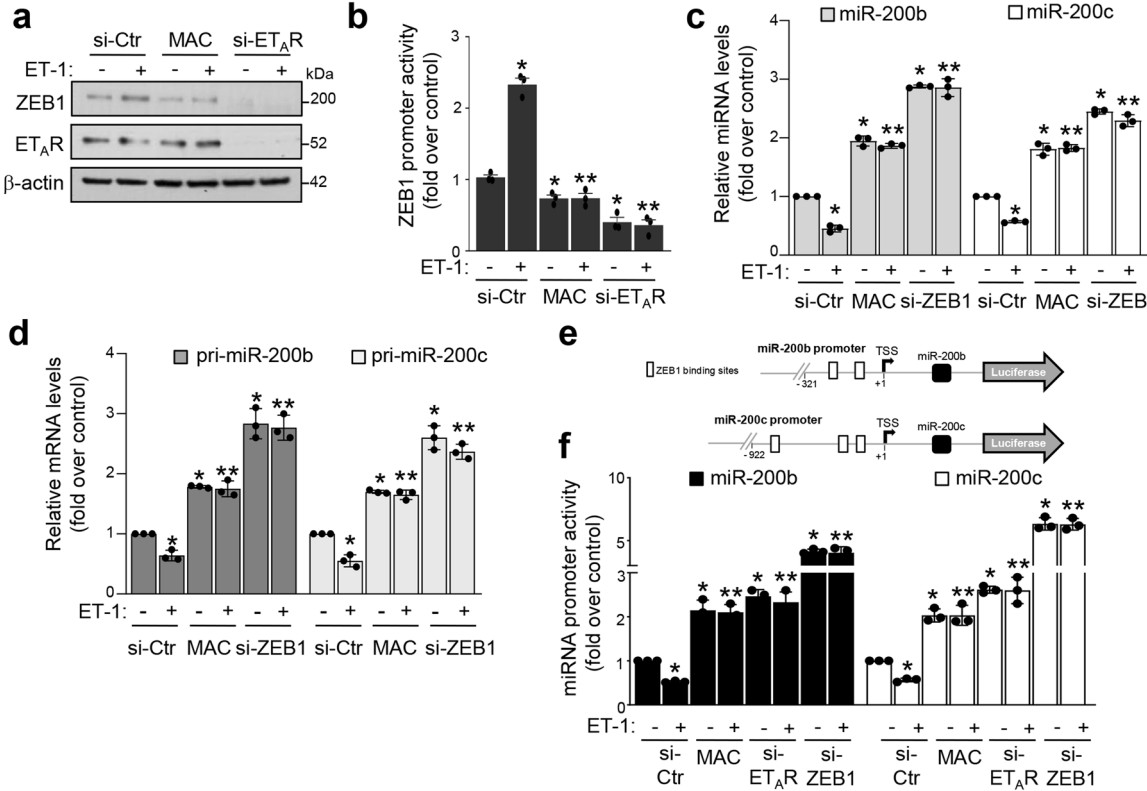

**Fig. 5 ET-1/ET$_A$R axis downregulates miR-200b/c via ZEB1. a** Expression of ZEB1 and ET$_A$R proteins in HEY cells transfected for 48 h with siRNA control (si-Ctr) or si-ET$_A$R or treated with MAC and stimulated or not for 48 h with ET-1 analyzed by WB. β-actin is used as loading control. **b** Luciferase activity in HEY cells co-transfected for 48 h with si-Ctr, si-ET$_A$R, and the ZEB1 promoter reporter plasmid and stimulated with ET-1 and/or MAC for 24 h. Values are the mean ± SD expressed as fold induction ($n = 3$; *$p < 0.0001$ vs. si-Ctr; **$p < 0.0001$ vs. ET-1 stimulated si-Ctr). **c** qRT-PCR for miR-200b/c expression in HEY cells transfected with si-Ctr or si-ZEB1 and stimulated with ET-1 for 72 h in the absence or in the presence of MAC. U6 is used to normalize. Values are the mean ± SD ($n = 3$; *$p < 0.001$ vs. unstimulated Ctr; **$p < 0.001$ vs. ET-1-stimulated Ctr). **d** Pri-miR-200b/c expression in cells transfected and treated as in **c** for 48 h is evaluated by qRT-PCR and normalized to cyclophilin-A. Values are the mean ± SD ($n = 3$; *, $p < 0.01$ vs. unstimulated Ctr; **$p < 0.001$ vs. ET-1-stimulated Ctr). **e** Schematic representation of the ZEB1 binding sites contained in the miR-200b/c promoter reporter plasmids. **f** Luciferase activity in HEY cells co-transfected for 48 h with si-Ctr, si-ET$_A$R, or si-ZEB1 and the reporter plasmids for miR-200b/c promoters and stimulated or not with ET-1 and/or MAC for 24 h. Values are the mean ± SD expressed as fold induction ($n = 3$; *$p < 0.001$ vs. unstimulated Ctr; **$p < 0.001$ vs. ET-1-stimulated Ctr).

Fig. 5a). Of note, the inhibitory effect of ET-1 on E-cadherin expression was paralleled by its effect on E-cadherin promoter activity that was curbed by macitentan, or either ET$_A$R or ZEB1 depletion and by overexpression of miR-200b/c (Fig. 6d). In parallel, ET-1/ET$_A$R axis downregulated miR-200b/c expression levels and this effect was hampered by ZEB1 or ET$_A$R silencing, or macitentan treatment (Supplementary Fig. 5b), indicating the cooperation with ET-1/ET$_A$R pathways and ZEB1/miR-200 axis in EMT-related effects. Next, to further understand the fine-tuning of ET-1 receptor blockade in the regulation of miR-200b/c/ZEB1 circuit, we combined macitentan with anti-miR-200b/c treatment, demonstrating that the effect of macitentan on ZEB1 and E-cadherin expression was still maintained in unstimulated and ET-1-stimulated cells (Supplementary Fig. 5c). Altogether, these findings argue for an integrated circuit between ET-1/ET$_A$R axis and ZEB1/miR-200 loop. Moreover, to assess the involvement of the ET$_A$R-miR-200b/c-ZEB1 circuit in the ET-1 capacity to regulate cell plasticity, we analyzed the ability of aggressive cells to form vascular-like structures in a process named vasculogenic mimicry[37]. ET-1-stimulated cells were able to organize themselves into a network of vascular tubules with a significant increase of the number of nodes, and of the tube length compared to unstimulated cells (Fig. 6e and Supplementary Fig. 5d). Interestingly, the depletion of ZEB1 or ET$_A$R, as well as miR-200b/c overexpression significantly disrupted the formation of

these structures, similarly to macitentan (Fig. 6e and Supplementary Fig. 5d), indicating an important role of the ET$_A$R-miR-200b/c-ZEB1 circuit in cell plasticity and matrix degradative enzyme activity induced by ET-1. In line with these results, a reduced cell invasion was observed when ZEB1 or ET$_A$R was depleted, and miR-200b/c were overexpressed, or upon macitentan treatment (Fig. 6f and Supplementary Fig. 5e), indicating an ET$_A$R-miR-200b/c-ZEB1 network involvement in the ET-1-driven tumor aggressiveness.

**Macitentan impairs metastatic spread by interfering with the ET$_A$R-miR-200b/c-ZEB1 circuit.** To evaluate the impact of the ET$_A$R-miR-200b/c-ZEB1 interplay on metastatic progression in vivo, we performed experiments using different ovarian cancer xenograft models. In particular, to analyze the effect of macitentan on the metastatic nodule formation, we intraperitoneally implanted in nude mice SKOV3 cells, or HG-SOC cells, OVCAR-3, that carry a hot spot missense TP53 mutation[38] and express ET$_A$R (Supplementary Fig. 6a). Moreover, because ZEB1 has been associated with chemoresistance in ovarian cancer[39,40], we analyzed the anti-metastatic effect of macitentan in cisplatinum sensitive (A2780) or resistant (A2780 CIS) xenografts that overexpressed ET$_A$R (Supplementary Fig. 6a). Macitentan inhibited the ET-1-induced expression of ZEB1 in OVCAR-3, A2780 cells, and even in A2780 CIS cells that, as expected, showed higher

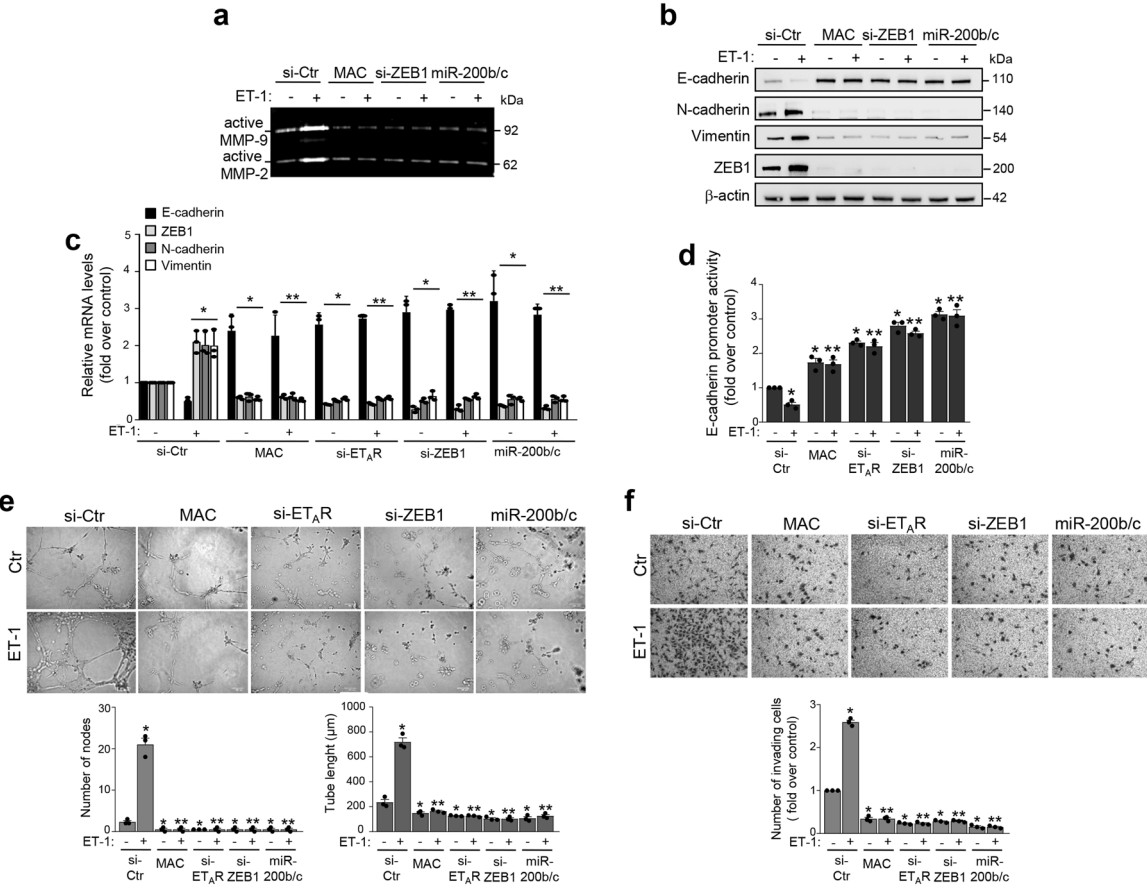

**Fig. 6 The integrated circuit ET$_A$R-miR-200b/c-ZEB1 is involved in ET-1-dependent protease activation, EMT, cell plasticity, and invasion.**
**a** Conditioned media from SKOV3 cells transfected with siRNA control (si-Ctr), si-ZEB1, or mimic-miR-200b/c (miR-200b/c), or treated with MAC, stimulated or not with ET-1 for 48 h are analyzed by gelatin zymography to detect the active forms of MMP-2/9. **b** Expression of E-cadherin, N-cadherin, Vimentin, and ZEB1 proteins is analyzed by WB in SKOV3 cells transfected and stimulated as in **a**. β-actin is used as loading control. **c** E-cadherin, ZEB1, N-cadherin, and Vimentin gene expression in SKOV3 cells transfected with si-Ctr, si-ZEB1, si-ET$_A$R, or mimic-miR-200b/c (miR-200b/c), and treated as in **b** is analyzed by qRT-PCR and normalized to cyclophilin-A. Values are the mean ± SD ($n = 3$; *$p < 0.02$ vs. si-Ctr; **$p < 0.004$ vs. ET-1 stimulated si-Ctr). **d** Luciferase activity in SKOV3 cells transfected as in **c** and co-transfected with the E-cadherin promoter reporter plasmid and stimulated with ET-1 and/or MAC for 48 h. Values are the mean ± SD expressed as fold induction ($n = 3$; *$p < 0.002$ vs. si-Ctr; **$p < 0.004$ vs. ET-1 stimulated si-Ctr). **e** Assay of tubule-like structure formation in SKOV3 cells transfected for 48 h as in **c** and overnight stimulated or not with ET-1. Original magnification 20×. (Scale bar: 100 μm). Graphs represent the quantification of the number of nodes and the tube length. Columns show the mean ± SD ($n = 3$; *$p < 0.02$ vs. unstimulated Ctr; **$p < 0.001$ vs. ET-1-stimulated Ctr). **f** Chemoinvasion assay in SKOV3 cells transfected as in **c** and overnight stimulated or not with ET-1. Images represent the crystal violet-stained invasive cells. Magnification ×10. Graph represents the number of invading cells. Columns show the mean ± SD ($n = 3$; *$p < 0.001$ vs. unstimulated Ctr; **$p < 0.001$ vs. ET-1-stimulated Ctr).

levels of ZEB1 (Supplementary Fig. 6b). Similarly, macitentan treatment hampered the ET-1 mediated reduction of miR-200b/c levels in these cells (Supplementary Fig. 6c). The ovarian cancer xenograft models treated with macitentan exhibited a reduced number of intraperitoneal nodules (Fig. 7a, b) associated with a well-tolerated toxicity profile (no weight loss in the treated mice). Of note, we observed a significant reduction of ZEB1 expression that was coupled with an inhibition of vimentin and an enhanced E-cadherin expression, at mRNA and protein levels, in the intraperitoneal metastatic nodules from macitentan-treated mice (Fig. 7c, d and Supplementary Fig. 6d). Notably, this effect was paired with the concomitant in vivo upregulation of miR-200b/c (Fig. 7e). Interestingly, ChIP assays on nuclear extracts from metastatic nodules of SKOV3 xenografts, revealed the recruitment of endogenous ZEB1 on miR-200b promoter. The treatment with macitentan impaired this recruitment in vivo, restraining the regulatory interaction of ZEB1 on the transcriptional activity of miR-200 (Fig. 7f). Altogether, these in vivo findings strengthen the existence of an ET$_A$R-miR-200b/c-ZEB1 regulatory circuit

able to control metastasis formation that can be hampered by macitentan.

## Discussion
Understanding of the dynamic properties of regulatory networks involved in the metastatic progression is crucial towards the improvement of the clinical outcome of ovarian cancer. The tumor aggressiveness is regulated by the complex changes in EMT-associated cell phenotypes, critically mediated by the EMT-TF as reflected by their association with poor clinical outcome in many tumors[41]. The many signaling cues that activate EMT-TF increase the complexity of the turning on of EMT program in tumor cells. In addition, EMT-TF are differentially regulated at the post-transcriptional level by miRNAs[8]. A prominent example of a mechanism that favors tumor aggressiveness is the reciprocal relationship between ZEB1 and miR-200 family members[18–20].

In an attempt to elucidate the mechanisms controlling ovarian cancer progression, in the present study we have

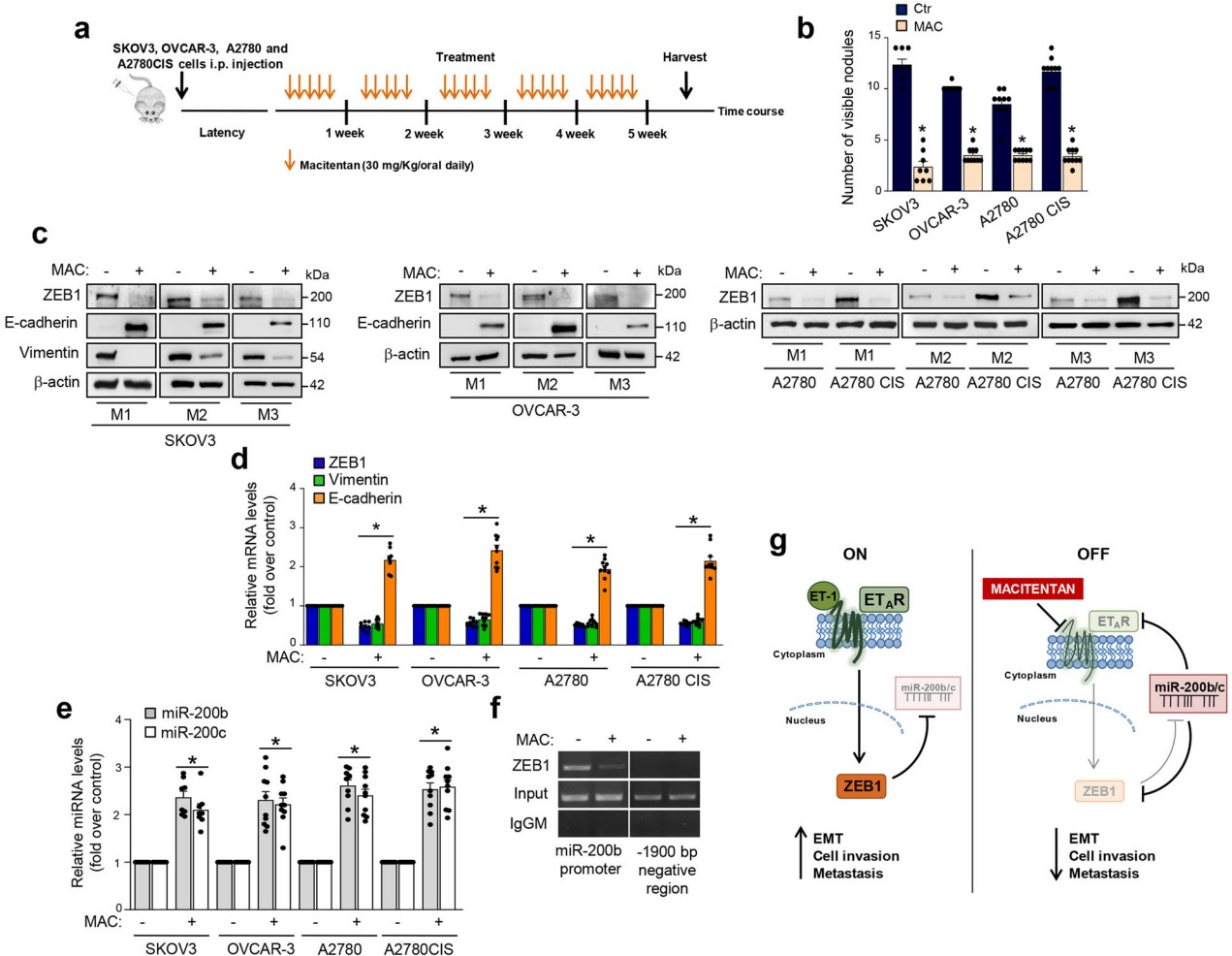

**Fig. 7 Macitentan impairs metastatic spread by interfering with the ET$_A$R-miR-200b/c-ZEB1 circuit. a, b** Female nude mice are i.p. injected with SKOV3, OVCAR-3, A2780, or A2780 CIS cells and treated with vehicle (Ctr) or MAC (30 mg/kg oral daily) for 5 weeks. Graph represents the number of visible metastasis. Columns show the mean ± SD (*$p < 0.001$ vs. Ctr). **c** WB analysis of the indicated proteins in lysates from i.p. nodules of three representative (M1-M3) SKOV3, OVCAR-3, A2780, and A2780 CIS xenografts treated as in **a**. β-actin is used as loading control. **d** ZEB1, Vimentin, and E-cadherin gene expression in i.p. nodules of SKOV3, OVCAR-3, A2780, and A2780 CIS xenografts treated as in **a** is analyzed by qRT-PCR and normalized to cyclophilin-A. Values are the mean ± SD (*$p < 0.004$ vs. Ctr). **e** miR-200b/c expression in i.p. nodules of SKOV3, OVCAR-3, A2780, and A2780 CIS xenografts treated as in **a** is analyzed by qRT-PCR. U6 is used to normalize. Values are the mean ± SD (*$p < 0.006$ vs. Ctr). **f** The binding of ZEB1 on miR-200b promoter region and on a region -1900 bp upstream the miR-200b TSS site (lacking ZEB1 binding sites, negative control for non-specific enrichment) is analyzed by ChIP assay followed by PCR in SKOV3 xenografts treated as in **a**. Anti-IgG mouse (IgGM) Ab is used as control for all ChIP reactions. **g** Schematic model of a regulatory circuit established by ET$_A$R-miR-200b/c-ZEB1 to control metastatic progression that is activated by ET-1 and inhibited by macitentan.

identified ET-1/ET$_A$R signaling in the regulation of the ZEB1/miR-200 circuit. Of clinical relevance, in HG-SOC patients, which accounts for 70–80% of ovarian cancer deaths, an ET$_A$R/ZEB1 signature is associated with poor prognosis, highlighting the worse outcomes generated by the integration between ET$_A$R and ZEB1 transcriptional machinery for these patients still suffering from limited treatment option. Following this discovery, we investigated the relationship between the ET$_A$R/ZEB1 axis and miRNA, identifying miR-200b/c as regulators of both ET$_A$R and ZEB1. How the miR-200 family members regulate ovarian cancer progression is not completely understood. Thus, studies of clinical samples are not conclusive to relate the impact of miR-200 expression with disease stage[42–46]. Several evidence suggest that these miR-200 family members might serve as prognostic markers for ovarian cancer clinical outcome[47–51] and, in particular, a multivariate analysis confirmed that downregulation of miR-200c is associated with overall survival, independent of other clinical covariates[47]. In addition, miR-200b/c are downregulated in most

of ovarian cancer patients classified at high risk of relapse compared with no-relapsers[33]. Conversely, Cao et al.[52] reported that elevated miR-200b/c expression was associated with advanced tumors. Here, we discover that miR-200b/c, besides targeting ZEB1, specifically bind the ET$_A$R mRNA and reduce its expression in ovarian cancer cells. As a consequence of ET$_A$R inhibition, forced expression of miR-200b/c abrogates the ET-1/ET$_A$R-driven aggressiveness-related functions. In line with these findings, gene expression analysis demonstrated that overexpression of miR-200b reduces ET$_A$R mRNA levels in ovarian cancer cells[53], and similarly miR-200c downregulates ET$_A$R expression in gastric and high metastatic nonsmall cell lung cancers[54,55]. The double-negative feedback loop between miR-200 family members and ZEB1 in ovarian carcinoma represents a molecular switch regulating the fine-tuning and reversibility of EMT[18]. However, the underlying mechanism of ZEB1/miR-200 axis activation in ovarian cancer progression is still not clear. Interestingly, Bendoraite et al.[22] reported that the ovarian surface of  mesothelial

cells, displaying mesenchymal and epithelial characteristics, initially acquires characteristics of epithelial cells switching from a miR-200 family$^{LOW}$ and ZEB1/2$^{HIGH}$ state to a miR-200 family$^{HIGH}$ and ZEB1/2$^{LOW}$ phenotype. Subsequently, EMT may occur at later stages of tumor progression. In line with these results, we found that ovarian cancer cells express miR-200b/c$^{LOW}$ and ZEB1$^{HIGH}$, along with ET$_A$R$^{HIGH}$ and E-cadherin$^{LOW}$, consistent with a mesenchymal cell phenotype, predominant in aggressive cancers[56]. In this framework, it is increasingly evident the existence of a network of factors being affected by, and affecting, the critical pathways that change the cell fate. Accordingly, we observed that the mutually exclusive relationship between miR-200b/c and ZEB1 is regulated by ET-1/ET$_A$R axis in mesenchymal ovarian cancer cells, in HG-SOC cells, that carry a hot spot missense *TP53* mutation, and in platinum-resistant cells. The current study, revealing a negative association between ET$_A$R/ZEB1 and miR-200b/c expression in HG-SOC patients, suggests that the deregulation of these miRNAs may contribute to the aberrant expression of ET$_A$R and ZEB1 in aggressive tumors. We corroborate the existence of the ZEB1/miR-200 feedback loop, demonstrating that ZEB1 suppresses miR-200b/c by inhibiting their transcription. In addition, we reveal that ET-1/ET$_A$R pathway is an integral signaling cue of this double ZEB1/miR-200 feedback loop, suggesting a more complex regulation pattern controlling cancer progression. In particular, ET-1/ET$_A$R axis, suppressing the transcription of miR-200b/c through the regulation of ZEB1, endorses a mechanism of self-reinforcing. These findings suggest that changes in the level of ET$_A$R/ZEB1 axis and miR-200b/c could be important to control mesenchymal states and the ovarian cancer outcome.

Of translational interest, targeting ET$_A$R activity in ovarian cancer cells and xenografts, by using the FDA approved small molecule macitentan, results in the inhibition of invasive behavior and metastatic progression with the concomitant reduction of ZEB1 and the increase of miR-200b/c expression, validating the existence of the integrated ET$_A$R-miR-200b/c-ZEB1 circuit in metastatic nodules. These findings contribute to identify the networks, which regulate the function of EMT-TF, like miR-200/ZEB1, adding the functional link with ET-1/ET$_A$R signaling pathway supporting the basis for the therapeutic way to interfere with the interrelated ZEB1-dependent mechanism of ovarian cancer progression.

As summarized in the working model in Fig. 7g, our findings demonstrate a critical tumor suppressor role of miR-200b/c in ovarian cancer through the regulation of the ET$_A$R pathway. Moreover, we unveil that the miR-200b/c-ET$_A$R axis is modulated by the transcription factor ZEB1, suggesting that the ET$_A$R-miR-200b/c-ZEB1 network may represent a potential therapeutic avenue to control metastatic progression, that could be impeded by ET-1R blockade. The present study shows that ET$_A$R/ZEB1 targeting by miR-200 heightens ET-1 signaling and consequently can generate a condition of ET$_A$R addiction in ovarian cancer cells. Using different approaches, we demonstrate that the ET$_A$R-miR-200b/c-ZEB1 integrated circuit acts as a key node to confer aggressive traits to ovarian cancer cells fostering tumor progression. The interruption of this circuit by using ET-1R antagonist increases miR-200b/c levels, which, in turn, repress both ET$_A$R and ZEB1, and impair metastatic progression.

Improved understanding of how pathways operate and interact in the complex network governing the EMT, and chemoresistance of ovarian cancer cells may provide new strategies to develop more effective treatments[57]. Considering that high levels of ET$_A$R[28], as well as ZEB1 overexpression[39,40], have been associated with chemoresistance and poor prognosis in HG-SOC, treatment with macitentan, able to interfere with the ET$_A$R-miR-200b/c-ZEB1 circuit and to exert antimetastatic effect, may represent a valuable therapeutic option in HG-SOC cells, and in platinum-resistant cells, in which this interdependent circuit could represent a strategy to escape chemotherapy response. Although we evidenced a specific role of ET-1/ET$_A$R activation as molecular determinant in controlling the ZEB1/miR-200b/c circuitry in the metastatic cascade, the dual ET-1R antagonists might be more appropriate than high specific ET$_A$R antagonists in a clinical setting[25,57,58]. Indeed, the therapeutic benefit of the dual ET$_A$R/ET$_B$R-antagonist is to target not only ovarian cancer cells expressing ET$_A$R, hampering the ZEB1/miR-200 network, but also to interfere with tumor microenvironmental elements, such as cancer-associated fibroblasts, blood, lymphatic, and immune cells, tumor-associated macrophages, which mainly expressed ET$_B$R[25,28,59–61] representing a therapeutic strategy which may be used to design targeted therapies that can impair ZEB1/miR-200 transcriptional machinery in cancer cells and the complex regulatory circuits in the stromal architecture.

Currently new drug combinations are investigated in order to improve therapeutic outcome of ovarian cancer patients, including the combination with antivascular endothelial growth factor therapy, bevacizumab[62]. Interestingly, treatment with bevacizumab influences ET-1 plasma level[63,64] and is of greater benefit for those ovarian cancer patients with the mesenchymal subtype[65] in which ET$_A$R and ZEB1 are overexpressed. Therefore, we can envisage that bevacizumab may be more effective in combination with ET-1R antagonist, curbing the ET$_A$R/miR-200/ZEB1 network in this molecular subtype. Future combination therapy preclinical studies should be conducted to design successful trials reflecting the full therapeutic potential of ET-1R antagonists in ovarian cancer treatment, targeting the miR-200/ZEB1 circuit.

In conclusion, our results highlight a multi-factorial feedback system controlled by ET-1 to favor cell plasticity and metastatic behavior of ovarian cancer cells, that involves ET$_A$R and ZEB1 and is regulated by miR-200 family members, thereby providing new insights for ovarian cancer diagnosis and effective therapeutic intervention.

## Methods

**Cells and cell culture conditions.** Human ovarian carcinoma cell lines OVCA 433 were kindly provided by Prof. G. Scambia (Catholic University School of Medicine, Rome, Italy). Human ovarian cancer cell lines CAOV3, SKOV3, HEY, and OVCAR-3, carrying the hot spot missense *TP53* mutation (R248Q), were obtained from the American Type Culture Collection (VA, USA) and A2780 and its cis-platinum resistant subclone, A2780 CIS, were obtained from European Collection of Cell Cultures (UK). To retain platinum resistance, 1 µmol/l cisplatin was added to the culture medium every two passages. HEY, OVCAR-3, A2780, and A2780 CIS cells were grown in RPMI-1640 medium (cat. #618700010, Gibco Thermo Fisher Scientific, MA, USA), SKOV3 cells were grown in McCoy's 5A medium (cat. #26600023, Gibco Thermo Fisher Scientific), whereas OVCA 433 and CAOV3 cells were grown in Dulbecco's modified Eagle Medium (cat. #21885025, Gibco Thermo Fisher Scientific). All media were supplemented with 10% fetal calf serum, 50 units/ml penicillin, and 50 mg/ml streptomycin. The medium for OVCAR-3 cells was also supplemented with 1× nonessential amino acid (MEM) (Gibco). Cells were tested routinely for cell proliferation as well as mycoplasma contamination. Cell lines were validated by short tandem repeat (STR) profiling. Before each experiment, cells were serum starved by incubation in serum-free medium for 24 h. ET-1 was used at 100 nM and was purchased from Sigma-Aldrich (Germany). Macitentan, also known as ACT-064992 or N-(5-[4-bromophenyl]-6-{2-[5-bromopyr-imidin-2-yloxy]-ethoxy}-pyrimidin-4-yl)-N′-propylsulfamide, was added 30 min before ET-1 at a dose of 1 µM and was kindly provided by Actelion Pharmaceuticals, Ltd. (Switzerland). ZD4054, N-(3-methoxy-5-methylpyrazin-2-yl)-2-(4-[1,3,4-oxadiazol-2-yl]phenyl) pyridine-3-sulfonamide, kindly provided by AstraZeneca (Italy), was added 30 min before ET-1 at a dose of 1 µM. BQ788 was added 30 min before ET-1 at a dose of 1 µM and was purchased from Peninsula Laboratories, Abbott (Park, IL).

**RNA isolation and quantitative real-time PCR (qRT-PCR).** Total RNA was extracted using the Trizol reagent (Life Technologies, Italy) according to the manufacturer's protocol. First-strand complementary DNA was synthesized using SuperScript VILO cDNA synthesis kit (Life Technologies). qRT-PCR was

performed using Power SYBR Green PCR Master Mix (Applied Biosystems, NJ, USA) with a 7500 Fast Real-Time PCR System (Applied Biosystems) according to the manufacturer's instructions. The mRNA expression levels were determined by normalizing to cyclophilin-A mRNA expression. The primers employed for qRT-PCR were as follows: $ET_AR$ Fw: 5′-GGGATCACCGTCCTCAACCT-3′; $ET_AR$ Rev: 5′-CAGGAATGGCCAGGATAAAGG-3′; ZEB1 Fw: 5′-GCAGTCCAAGAACCAC CCTT-3′; ZEB1 Rev: 5′-GGGCGGTGTAGAATCAGAGT-3′; pri-miR-200b Fw: 5′-AGCAGCTCCTGGAAC-3′; pri-miR-200b Rev: 5′-CACGTGCTGCCTTGT-3′; pri-miR-200c Fw: 5′-AGCCAGGGATCTGCA-3′; pri-miR-200c Rev: 5′-ACCTTG GGTCAGGCA-3′; cyclophilin-A Fw: 5′-TTCATCTGCACTG CCAAGAC-3′; cyclophilin-A Rev: 5′-TCGAGTTGTCCACAGTCAGC-3′. For miRNA analysis RNA was reverse transcribed using hsa-miR-141, hsa-miR-200a, hsa-miR-200b, hsa-miR-200c, hsa-miR-429, or U6 snRNATaqMan MicroRNA Assay systems (Applied Biosystems) and qRT-PCR was performed by using Kapa Probe Fast Master Mix (KapaBiosystems, MA, USA). miRNA levels were normalized to U6 snRNA. Final data were obtained by using $2^{-\Delta\Delta Ct}$ method.

**Western blot analysis.** Cells were lysed in modified RIPA buffer (50 mM Tris-HCl pH 7.4, 250 mM NaCl, 1% Triton X-100, 1% sodium deoxycholate, and 0.1% SDS) containing a mixture of protease and phosphatase inhibitors. Protein content of the extracts was determined using the Bradford assay (cat. #5000001, Bio-Rad, CA, USA). Total proteins were subjected to SDS-PAGE. Denatured samples were loaded onto a gel of polyacrylamide and transferred by using Trans-Blot transfer pack (Bio-Rad). The membranes were blocked in TTBS (TBS with 0.1% Tween 20) containing either 5% dry milk or BSA. Primary antibodies (Abs) incubations were performed in TTBS with either 5% dry milk or BSA overnight at 4 °C. The Abs used for this study were as follows: anti-$ET_AR$ (1:3000, ab117521, Abcam, UK), anti-Flag-tag-DYKDDDDK (1:1000, cat. #2368, Cell Signaling Technology, MA, USA), anti-vimentin (1:1000, clone D21H3, cat. #5741, Cell Signaling Technology), anti-TCF8/ZEB1 (1:1000, clone D80D3, cat. #3396, Cell Signaling Technology), anti-E-cadherin (1:500, clone 36, cat. #610181, BD Biosciences, Belgium), anti-N-cadherin (1:500, clone 32, cat. #610921, BD Biosciences), and anti-β-actin (1:1000, clone I-19, cat. #sc-1616, Santa Cruz Biotechnology, CA, USA). After washing, membranes were incubated with the appropriate secondary Abs. Western blotting (WB) filters were analyzed using Clarity ECL detection system (cat. #1705061, Bio-Rad) or LiteAblot turbo extrasensitive chemiluminescent substrate (cat. #EMP012001, Euroclone, Italy). WB signals were quantified using ImageJ software[66].

**Transient transfection.** HEY and SKOV3 cells were transiently transfected with *mir*Vana miRNA mimic-hsa-miR-200b-3p (miR-200b), mimic-hsa-miR-200c-3p (miR-200c), negative miRNA control #1 (miR-Ctr), miRNA inhibitor hsa-miR-200b-3p (anti-miR-200b), miRNA inhibitor hsa-miR-200c-3p (anti-miR-200c), miRNA inhibitor negative control #1 (anti-miR-Ctr) (ThermoFisher Scientific, MA, USA) or with miScript mimic-miR-141, mimic-miR-200a, mimic-miR-429, or mimic-miR negative control (Qiagen, Germany) at a final concentration of 10–50 nM using RNAi/MAX Lipofectamine (ThermoFisher Scientific), according to the manufacturer's suggestions. Alternatively, cells were transfected with Dharmacon Smart Pool ON-TARGETplus siRNA oligonucleotides specific for EDNRA (si-$ET_AR$, L-005485–00–0020) or ZEB1 (si-ZEB1, L-006564–01–0020) or with ON-TARGETplus nontargeting pool (si-Ctr, D-001810–10–05) (GE Healthcare Life Sciences, MA, USA). Otherwise, cells were transiently transfected with 0.5–1 µg of the following expression plasmids: $ET_AR$ ORF clone expression plasmid ($ET_AR$ w/o the 3′UTR, cat. #RG205385, Origene, MD, USA) that only contains the coding sequence of $ET_AR$; wt $ET_AR$ expression plasmid, that has been synthesized by Neo Biotech (France) by adding the 3′UTR to the $ET_AR$ ORF clone plasmid; ZEB1 expression plasmid or DB-ZEB1-Flag (carrying a sequence able to block the ZEB1 DNA binding domain), both kindly provided by D. Dean[34]. C-2 pEGFP plasmid was used as control (Mock, Takara, CA, USA) and Lipofectamine 2000 (Life Technologies) as transfecting agent, according to the manufacturer's instructions.

**Luciferase reporter gene assay.** Luciferase assays were carried out in HEY and SKOV3 cells seeded in 12-well plates and transfected with 200–500 ng of reporter plasmids using Lipofectamine 2000 (Life Technologies), according to manufacturer's instructions. ZEB1 3′UTR reporter plasmid (SC-219122, Origene), $ET_AR$ 3′UTR reporter plasmid (SC-218505, Origene) and Mut$ET_AR$ 3′UTR plasmid, carrying a double mutation in positions 498–500 and 1726–1728, synthesized by Blue Heron Biotech (WA, USA). Δ$ET_AR$ 3′UTR plasmid, carrying a deletion of the region 1781–1784, was generated by using the QuikChange Site-Directed Mutagenesis Kit (Stratagene, CA, USA) and the following primers: Fw: 5′-GGAGCAAAAGTCATTA CACTTTGAATATATTGTTCTTATCCTCAATTCAA-3′; Rev: 5′-TTGAATTGAG GATAAGAACAATATATTCAAAGTGTAATGACTTTTGCTCC-3′. mut/Δ$ET_AR$ 3′UTR was generated by deleting the region between 1781 and 1784 bp in the mut $ET_AR$ 3′UTR plasmid using the following primers: Fw: 5′-TTGAATTGAGGATAA GAACAATATATTCAAAGTGTAATGACTTTTGCTCC-3′; Rev: 5′-GGAGCAAAA GTCATTACACTTTGAATATATTGTTCTTATCCTCAATTCAA-3′. miR-200 promoter activities were studied by using the pGL3–321/+120 construct (cat. #35540, Addgene, MA, USA), containing a region of 321 bp upstream to the Transcription Start Site (TSS) of miR-200b[21] and the 0.9 kb hmiR-200c promoter reporter,

containing a region of 922 bp upstream to the TSS of miR-200c, kindly provided by Dr. Muneesh Tewari[22]. ZEB1 transcription was analyzed by employing a luciferase reporter plasmid containing a 900 bp sequence from the ZEB1 promoter[67] synthesized by TEMA Ricerca (Italy). To analyze E-cadherin promoter activity we used the pGL2-Ecad3 construct, kindly provided by Dr. E.R. Fearon (University of Michigan, Ann Arbor, MI, USA). All plasmids were co-transfected with 150 ng of pCMV-β-galactosidase vector (Promega) and 20–50 nM of siRNAs, mimic-miRNAs or anti-miRNAs as indicated. Reporter activity was measured using the Luciferase assay system (Promega) and normalized to β-galactosidase activity.

**Cell viability analysis.** HEY cells were seeded in 12-well plates in triplicate. After 24 h cells were transfected and stimulated as indicated for 48–72 h. Then, cells were harvested and counted by using trypan blue dye exclusion method.

**Gelatin zimography.** To detect MMP-2/9 activity conditioned media of SKOV3 cells were collected and concentrated by using Spin-X UF concentrator columns (Corning, NY, USA). Samples were separated by 9% SDS/PAGE gels containing 1 mg/ml gelatin. The gels were washed for 30 min at 22 °C in 2.5% Triton X-100 and then incubated in 50 mMTris (pH 7.6), 1 mM $ZnCl_2$, and 5 mM $CaCl_2$ for 18 h at 37 °C. After incubation gels were stained with 0.2% Coomassie Blue. Enzyme-digested regions were identified as white bands on a blue background and images were acquired with ChemiDoc Imaging System (Bio-Rad).

**Transwell invasion assay.** Invasion assays were carried out using Boyden Chambers consisting of transwell filter inserts with 8 µm size polycarbonate membrane (Corning) placed in a 24-well plate and precoated with polymerized collagen (BD Biosciences). Transfected SKOV3 and HEY cells ($3 \times 10^4$) were seeded with serum-free medium in the upper chamber and ET-1 added or not to the lower chamber. Cells were left to invade overnight at 37 °C. Cells on the upper part of the membrane were scraped using a cotton swab and the migrated or invaded cells were stained using Diff-Quick kit (Merz-Dade, Switzerland). From every transwell, several images were taken under a phase-contrast with Olympus I×70 microscope (Olympus Corporation, Japan) at 10× magnification and two broad fields were considered for quantification.

**Vasculogenic mimicry assay.** SKOV3 and HEY cells ($2 \times 10^4$) were seeded in a 96-well culture plate precoated with 50 µl/well of growth factor reduced Cultrex (Trevigen) and stimulated with ET-1 or MAC for 16 h. Upon treatment, tubule-like structure formations were visualized with an inverted microscope with a 20× magnification. Representative images were captured with a ZOE Fluorescent Cell Imager (BioRad Laboratories). Tube formation was analyzed by using Angiogenesis Analyzer for ImageJ (NIH) measuring the number of nodes and the tube length.

**Animal study.** Female athymic (nu+/nu+) mice, 4–6 weeks of age (Charles River Laboratories, Italy) were injected intraperitoneally with $2.5 \times 10^6$ viable SKOV3, OVCAR-3, A2780, and A2780 CIS cells following the guidelines for animal experimentation of the Italian Ministry of Health. Two weeks after, SKOV3 xenografts were randomized into two different groups of eight mice, whereas OVCAR-3, A2780, and A2780 CIS xenografted mice into two different groups of ten mice undergoing the following treatments: control (Ctr; vehicle) vs. macitentan (MAC; 30 mg/kg/oral daily) for 5 weeks. After the end of treatment, all mice were euthanized and intraperitoneal organs were analyzed. The number of visible metastases was counted and the removed i.p. nodules were measured, carefully dissected, frozen and processed for WB and qRT-PCR analyses. Values represent the mean ± SD of eight mice for group for SKOV3 xenografts and ten mice for group for OVCAR-3, A2780, and A2780 CIS xenografts from two independent experiments.

**Chromatin immunoprecipitation (ChIP).** Chromatin was extracted from SKOV3 cell lines ($5 \times 10^6$) or from SKOV3 xenografts and ChIP assays were performed as previously described[28]. Chromatin was sheared by sonication, centrifuged and diluted in 50 mM Tris pH 8.0, 0.5% NP-40, 0.2 M NaCl, and 0.5 mM EDTA. One-twentieth of the precleared chromatin was used as the input for the ChIP assay. The precleared chromatin was rotated overnight with: anti-ZEB1 (2 µg/µl, clone H3, sc-515797, Santa Cruz Biotechnology, CA, USA) and antimouse IgG Isotype Control (2 µg/µl, Invitrogen, Carlsbad, California, USA). The differential binding between proteins and promoters DNA was examined by PCR. The primers used were as follows: miR-200b promoter, 5′-CTGCGTCACCGTCACTGG-3′ and 5′-ACAACTCGCCCGTCTCTG-3′; −1900 bp negative region, 5′-CAGCAGGTTT TCCACCACAG-3′ and 5′-GAAGCTGCTCTTTCTCCAAGG-3′.

**Bioinformatic analysis.** Normalized miRNA and gene expression of HG-SOC were obtained from Broad Institute TCGA Genome Data Analysis Center (2016): TCGA data from Broad GDAC Firehose 2016_01_28 run. Broad Institute of MIT and Harvard. Dataset (https://doi.org/10.7908/C11G0KM9). All the expression value differences between subgroups of samples were evaluated applying unpaired *t*-test or permutation test where specified. Pearson's correlation coefficient was calculated between miRNA/gene target or gene/gene expression. One-way analysis

of variance (ANOVA) was employed to analyze differences of gene expression between the subgroups of patients from TCGA. Survival and progression-free survival were evaluated by Kaplan–Meier method and a log-rank test was used to establish the statistical significance of the distance between curves. Analysis of data was performed by using the MATLAB software (MA, USA).

**Statistics and reproducibility**. All experiments were repeated at least three times with comparable results, unless indicated otherwise. Exact sample sizes and number of replicates were indicated in the figure legends. Results are expressed as mean ± standard deviation (SD). Error bars in bar charts represent SD. Statistical analyses were performed by using the GraphPad Prism version 8.0.0 software (CA, USA, www.graphpad.com). Student's $t$-test was employed to analyze the in vitro and in vivo data. A threshold $p < 0.05$ was defined as statistical significance.

**Reporting summary**. Further information on research design is available in the Nature Research Reporting Summary linked to this article.

## Data availability
The source data underlying the graphs presented in the main figures are shown as Supplementary data. Uncropped blots of major figures are shown in Supplementary Fig. 7. All other data supporting the findings of the study are available within the paper and Supplementary information.

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

## Acknowledgements

We gratefully acknowledge Aldo Lupo for technical assistance and Maria Vincenza Sarcone for secretarial support. This work was supported by Associazione Italiana Ricerca sul Cancro (AIRC) to A.B. (AIRC IG22835).

## Author contributions

A.B. conceived and supervised the project. A.B. and R.S. analyzed and discussed the data with the help of G.B. R.S. performed most of the experiments shown in this work with the help of R.C., P.T., and L.R. A.S. performed the bioinformatic analyses. A.B. and R.S. wrote the paper with input from the other authors. All authors provided comments.

## Competing interests

The authors declare no competing interests.
