## [Peer Review File · Communications Biology]

Reviewers' comments:

Reviewer #1 (Remarks to the Author):

In this manuscript, the authors propose and characterize a complex network integrating ET-1/ETAR and the miR-200b/c-Zeb1 feedback loop, which they suggest to be highly relevant in ovarian cancer metastatic progression. The pivotal role of the miR-200b/c-Zeb1 feedback loop in epithelial cancers has been extensively studied. In this study the authors demonstrate an association between ETAR and Zeb1 expression and disease progression in ovarian cancer. The inter-regulation of the members of this feedback loop is carefully characterized via numerous in vitro experiments, utilizing various tools to manipulate expression and block activity of the different members. Moreover, functional assays and in vivo xenografts are performed to suggest potential disease relevance of this axis. While the existence and relevance of the proposed network seems likely, the authors frequently draw conclusions from data with insufficient replicates. In order to strengthen this study, it is thus necessary that more replicates and quantification (notably of western blots) be included. In addition, more careful interpretation and description of the data would be helpful for the reader to understand a very complex regulatory network.

Major concerns:

1. The mechanism of how the ET-1/ETAR signaling positively regulates ZEB1 expression (transcriptional, translational control?) remains unknown.
2. Western blots throughout the paper frequently only show one replicate per treatment. While this is fine for representative purposes, the authors should include blots with more replicates in the supplementary, along with quantification, especially for experiments where no additional data is present to support the findings (qPCR, luciferase assay, etc.), or if bands are weak or unclear. This is a concern for example for Fig. 6b, which has no further data for support, and for which some changes seem subtle. It is most notable in Fig 7, in which a large animal experiment was performed, but the only expression data shown for EMT genes is 1 vs. 1 in Fig 7b. The text states that this is a "significant reduction of ZEB1 expression," which is impossible to conclude based on the provided data. The authors should include western blots with more replicates and potentially also perform qPCR analysis of these genes using the RNA extracted to generate Fig 7c.
3. The data presented in Figure 6b are difficult to interpret. The authors claim that Zeb1 depletion inhibits ET1-induced changes in expression of E-cadherin, N-cadherin, and Vimentin. They also claim that miR-200b/c overexpression inhibits ET-1-dependent ZEB1 upregulation. While both of these statements appear true, it seems that the changes in expression induced by Zeb1-KD or miR-200b/c overexpression alone have a much greater effect than ET-1 treatment in the si-Ctr conditions, suggesting that the mechanisms for these conditions may be ET-1/ETAR-independent (and Zeb1 is known to repress E-cadherin expression directly, while miR-200 is a very strong repressor of Zeb1, as seen in Fig. 3c). What is more striking, however, is that MAC treatment induces such a strong upregulation of E-cadherin. Presumably this is due to reduction in expression of Zeb1, based on the results in Fig 7 (although this is not evident in Fig. 6b). To be able to better interpret these results, the authors should not only increase western blot replicates and provide quantification, but also show miR-200b/c expression levels. It might also be interesting to combine MAC treatment with mild Zeb1 overexpression or anti-miR treatment to see if the effects are still maintained. Furthermore, it would be helpful for the reader if the text is revised to account for these caveats or to better explain why they may not be of concern.
4. The miR-200b, c, 429 subfamily is genomically and transcriptionally linked to subfamily miR-141, 200a. The authors should comment on the role of this family on the ETAR - miR-200b/c-Zeb1 feedback loop. Have they tested this?

Minor concerns:

1. Text should be carefully revised for errors, as there are sometimes words missing, making it difficult for the reader to follow, especially given the complexity of the regulatory network.
2. In Fig. 2h and 2i, the authors overexpress ETAR lacking its 3'UTR, and state that miR-200b and miR-200c mimics don't reduce its expression. This could also be due to the strong overexpression of the truncated ETAR and a potential lack of sufficient copies of miRNA to affect its expression; the best control would be to compare with similar overexpression of WT ETAR, and show that the miR-200b and miR-200c mimics DO reduce overexpressed ETAR in this case.
3. Bottom of page 6: The authors state: "Unlike for the mutated ETAR 3'UTR reporter plasmid, ectopic miR-200b/c significantly reduced the ETAR reporter activity thereby confirming its direct targeting". This should be worded more carefully since the authors show the potential of the ETAR 3'UTR to be targeted by miR-200b/c, but no evidence is provided that this happens in OC in vivo. For this they would have to demonstrate miR-200/ ETAR 3'UTR duplexes loaded into Ago by AgoIP.

Reviewer #2 (Remarks to the Author):

In this work by the Bagnato laboratory, the authors present data suggesting the existence of a regulatory circuit comprising the endothelin A receptor, ZEB1, and miR-200b/c. The role of miR-200 family members in regulating ZEB1 in ovarian cancer through EMT has been well-known for more than a decade (Park SM *Genes Dev.* 2008; Bendoraite A *Gynecol Oncol* 2010). The authors have previously shown that miR-30a can also regulate EMT in ovarian cancer but through the endothelin-1 axis. Here, the authors describe cross-talk of endothelin with miR-200. This role for endothelin A in the regulation of miR-200b/c is new. Double-negative feedback loops have been described in multiple cell types, including ovarian cancer, with regards to EMT and stemness, so the existence of such a mechanism in ovarian cancer cells makes sense. Understanding these complex regulatory networks is very important in understanding this disease.

Specific Comments:

- 1) It is interesting that the authors suggest increased miR-200b/c expression and ZEB1 inhibition may be a therapeutic strategy. Others have also noted that loss of miR-200c is a marker of clinical aggressiveness and that restoration of miR-200c to aggressive cancer cells causes a decrease in migration and invasion (Cochrane DR *J Oncol* 2010) or that knockdown of ZEB1 slows tumor growth and metastasis (Chen D *Int J Gynecol Cancer* 2013). However, this is paradoxical as miR-200 family members are expressed at low or negligible levels in normal cells and substantially increase in expression in ovarian cancer, whereas expression of ZEB1 and ZEB2 shows the opposite pattern (Bendoraite A *Gynecol Oncol* 2010). How might the authors reconcile these roles for low miR-200 as a marker for both normal tissue and for aggressive cancer but not for the transition phase?
- 2) The authors chose to focus on SKOV3 and HEY 8A cells (here denoted simply as HEY). From a molecular biology perspective, this is fine as they express Vimentin and might be thought of as more mesenchymal cell lines. I think from the vantage point of illustrating a signaling pathway, cell line choice is less important. The authors should give the caveat, however, that the fidelity of these lines to high-grade serous ovarian cancer is low, as neither cell line has a tp53 mutation, and they likely represent a non-serous histology. The correlation with TCGA helps a little bit in this regard, but testing their hypotheses in a tp53 mutated line would give a better sense of the translational applications for targeting ZEB1 as a therapeutic approach. Is this response limited to

mesenchymal type cells? Either way, the in vivo experiments are limited to a single cell line. Performing a set of in vivo experiments with a tp53 mutated cell line would be informative whether or not a response was seen.

3) In a similar vein regarding translational utility, ZEB1 has been associated with chemoresistance in ovarian cancer (Sakata J Oncotarget 2017; Cui Y Chemotherapy 2018). What is the potential role of macitentan in chemoresistant ovarian cancer or in reversing chemoresistance?

4) In the discussion, the authors should note that endothelin-1 is a secondary target for bevacizumab. Supporting the authors hypothesis of endothelin-1 as a therapeutic target in mesenchymal subtype ovarian cancer, the mesenchymal subtype derives relatively greater benefit from bevacizumab than immunoreactive or differentiated subtypes.

Reviewer #3 (Remarks to the Author):

The manuscript shows relevant evidence to understand part of the mechanisms directing the EMT and how they are related to the OC progression. This work is based on previous findings in which authors determined a role of the ET-1/ETaR axis in EMT at OC. Currently, they present new data demonstrating the interaction of this axis with the transcription factor ZEB1, another regulator of EMT.

Despite authors determined the binding and regulatory potential of miR200b/c on ZEB1, the experiments are especially designed to define such interactions, as reporter assay for UTR sequences. Therefore, it may be discussed whether such interactions are relevant in an in vivo context with endogenous proteins. It should be interesting to evaluate the latter by CHIP assay in primary cultures and/or CO cell lines.

In Figure 4 authors show that ZEB1 regulates ETaR through repressing miR200b/c. The experiments are right and clearly show the regulatory feedback for expression of both proteins, ETaR and ZEB1, as well as miR200b/c. However, due to experimental conditions, it is necessary to establish the in vivo significance of the suggested mechanism considering the endogenous components. Authors should also discuss the latter due to the complex nature of signaling pathways and their diversity related to EMT in which an unique miR can have many has potential effects.

In figure 6 the role of ETaR-miR200b/c-ZEB1 pathway in ET-1 activation via EMT, cell plasticity and invasion are quite clear. However, it is unclear why authors did not use ETaR silencing, as indeed previously used (eg. Figs 2g and 5a). Why did they just perform pharmacological inhibition (macitentan) of ETaR?

Concluding with a glimpse of a therapeutic strategy derived from the findings is quite welcome. However, authors should discuss about possible pharmacological alternatives with higher specificity for ETaR, especially decreasing a potential side effect of macitentan. In addition, authors should discuss the extent that ETbR may have a role in the suggested mechanism.

Reviewer #1

Comment 1. The mechanism of how the ET-1/ETAR signaling positively regulates ZEB1 expression (transcriptional, translational control?) remains unknown.

Response: We thank the reviewer for these thoughtful and constructive comments pointing out how ET-1/ET_AR signaling control ZEB1 expression. To pursue it we assessed ZEB1 promoter activity (new Fig. 5b and new Suppl. Fig. 4e) upon ET-1 stimulation either in the presence or in the absence of macitentan (MAC), and in cells depleted for ET_AR expression. Of note, in HEY and SK-OV3 cells, in which we observed an ET-1-induced ZEB1 expression at protein (Fig 5a and new Suppl. Fig. 4c) and mRNA (Suppl. Fig. 4a) levels, ET-1 promoted ZEB1 promoter activity whereas MAC treatment, and similarly ET_AR depletion, impaired ET-1-induced ZEB1 promoter activity, thereby suggesting that ET-1/ET_AR axis positively regulates ZEB1 expression at the transcriptional level.

Comment 2. Western blots throughout the paper frequently only show one replicate per treatment. While this is fine for representative purposes, the authors should include blots with more replicates in the supplementary, along with quantification, especially for experiments where no additional data is present to support the findings (qPCR, luciferase assay, etc.), or if bands are weak or unclear. This is a concern for example for Fig. 6b, which has no further data for support, and for which some changes seem subtle. It is most notable in Fig 7, in which a large animal experiment was performed, but the only expression data shown for EMT genes is 1 vs. 1 in Fig 7b. The text states that this is a “significant reduction of ZEB1 expression,” which is impossible to conclude based on the provided data. The authors should include western blots with more replicates and potentially also perform qPCR analysis of these genes using the RNA extracted to generate Fig 7c.

Response: We thank the reviewer for this insightful comment. In fact, in the revised version of the manuscript, we included replicates and quantifications. Regarding Fig. 6b, we included new blots along with the replicate and related quantification in the new supplementary figures (new Fig. 6b and new Suppl. Fig. 5a). For some experiments we also included additional experimental approaches to further support our findings (qPCR, luciferase assay). In parallel, we provided further evidence showing the levels of EMT marker and miR200b/c expression (new Fig. 6c and new Suppl.Fig.5b) along with E-cadherin promoter activity (new Fig.6d), to better support our conclusions. Similarly, the new blots of the new Fig.7b more clearly shows more clearly than previously the reduction of ZEB1 induced by the treatment with MAC in all the in vivo set of xenograft tissues in 3 different mice. In addition, as requested, the new fig. 7c show the effect of MAC treatment also on the expression of EMT markers at mRNA level, as analyzed by qPCR analysis.

Comment 3. The data presented in Figure 6b are difficult to interpret. The authors claim that Zeb1 depletion inhibits ET1-induced changes in expression of E-cadherin, N-cadherin, and Vimentin. They also claim that miR-200b/c overexpression inhibits ET-1-dependent ZEB1 upregulation. While both of these statements appear true, it seems that the changes in expression induced by Zeb1-KD or miR-200b/c overexpression alone have a much greater effect than ET-1 treatment in the si-Ctr conditions, suggesting that the mechanisms for these conditions may be ET-1/ETAR-independent (and Zeb1 is known to repress E-cadherin expression directly, while miR-200 is a very strong repressor of Zeb1, as seen in Fig. 3c). What is more striking, however, is that MAC treatment induces such a strong upregulation of E-cadherin. Presumably this is due to reduction in expression of Zeb1, based on the results in Fig 7 (although this is not evident in Fig. 6b). To be able to better interpret these results, the authors should not only increase western blot replicates and provide quantification, but also show miR-200b/c expression levels. It might also be interesting to combine MAC treatment

with mild Zeb1 overexpression or anti-miR treatment to see if the effects are still maintained. Furthermore, it would be helpful for the reader if the text is revised to account for these caveats or to better explain why they may not be of concern.

Response: As mentioned above, we improved the quality of the blots (new Fig. 6b and new Suppl. Fig. 5a), along with relative quantifications, and qPCR analysis of EMT genes (new Fig. 6c) have been included in the revised manuscript. The new figures show better than the previous ones the effect of ET-1 on ZEB1 and EMT marker expression that was hampered upon macitentan treatment, as well as by depletion of ZEB1 and by overexpression of miR-200b/c. Of note, the effect of ET-1 on E-cadherin expression was paralleled by its effect on E-cadherin promoter activity that was curbed by macitentan, by either ET_AR or ZEB1 depletion, and by overexpression of miR-200b/c (Fig.6d). In parallel, ET-1/ET_AR axis downregulated miR-200b/c expression levels. This effect was hampered by ZEB1 or ET_AR silencing, or macitentan treatment (new Suppl. Fig. 5b) indicating the cooperation with ET-1/ET_AR pathways and ZEB1/miR-200 axis in EMT-related effects. As suggested by the reviewer, to further understand the fine-tuning of ET-1R blockade in the regulation of miR-200b/c/ZEB1 circuit, we combined macitentan with anti-miR-200b/c treatment (new Suppl. Fig. 5c). We found that the inhibitory effect of macitentan on ZEB-1 overexpression was still maintained in unstimulated and ET-1-stimulated cells. Altogether, these findings argue for an integrated circuit between ET-1/ET_AR axis and ZEB1/miR-200 loop. All these data are now discussed and better explained in the Results section text, to further highlight the importance of our findings describing the complexity of this ET-1/ET_AR-dependent novel regulatory network.

Comment 4: The miR-200b, c, 429 subfamily is genomically and transcriptionally linked to subfamily miR-141, 200a. The authors should comment on the role of this family on the ETAR – miR-200b/c – Zeb1 feedback loop. Have they tested this?

Response: Based on this suggestion, we evaluated the expression of the other members of miR200 family in all the analyzed ovarian cancer cell lines utilized. As showed in the new Suppl. Fig. 2b, all the OC cell lines analysed express the five miR-200 family members. We evaluated the effects of these miRNA on ET_AR expression and, as predicted by bioinformatic analysis, only miR-429 is able to target ET_AR. A negative association between miR-429 and ET_AR expression was observed in OC cell lines. The miR-200a and miR-141, able to target ZEB1, have no detectable effect on ET_AR expression (new Suppl. Fig. 2c), suggesting that miR-200b/c, along with miR-429, can regulate both ET_AR and ZEB1 expression.

Minor concerns:

1. Text should be carefully revised for errors, as there are sometimes words missing, making it difficult for the reader to follow, especially given the complexity of the regulatory network.
Response 1: Thanks for pointing this out. We revised the text to better introduce the topic and the potential importance of our findings describing the complexity of this novel regulatory network. The entire manuscript has been proofread and edited for grammatical and spelling mistakes.

2. In Fig. 2h and 2i, the authors overexpress ET_AR lacking its 3'UTR, and state that miR-200b and miR-200c mimics don't reduce its expression. This could also be due to the strong overexpression of the truncated ET_AR and a potential lack of sufficient copies of miRNA to affect its expression; the best control would be to compare with similar overexpression of WT ET_AR, and show that the miR-200b and miR-200c mimics DO reduce overexpressed ETAR in this case.

Response 2: We agree with the reviewer that the statement that miR-200b and miR-200c mimics don't reduce overexpressed ET_AR expression is not correct. As suggested by the reviewer in the next minor point, we performed a new experiment by using the best control comparing with similar overexpression of WT ET_AR (new Fig. 2h and new 2i), and showed that the miR-200b and miR-200c mimics reduce overexpressed ET_AR and cell vitality also in this case.

3. Bottom of page 6: The authors state: "Unlike for the mutated ETAR 3'UTR reporter plasmid, ectopic miR-200b/c significantly reduced the ETAR reporter activity thereby confirming its direct targeting". This should be worded more carefully since the authors show the potential of the ETAR 3'UTR to be targeted by miR-200b/c, but no evidence is provided that this happens in OC in vivo. For this they would have to demonstrate miR-200/ ETAR 3'UTR duplexes loaded into Ago by AgoIP.

Response 3: Based on this suggestion, we reworded the sentence with "ectopic miR-200b/c significantly reduced the ET_AR reporter activity thereby indicating the potential of the ET_AR 3'UTR to be targeted by miR-200b/c".

Reviewer #2

Comment 1: It is interesting that the authors suggest increased miR-200b/c expression and ZEB1 inhibition may be a therapeutic strategy. Others have also noted that loss of miR-200c is a marker of clinical aggressiveness and that restoration of miR-200c to aggressive cancer cells causes a decrease in migration and invasion (Cochrane DR J Oncol 2010) or that knockdown of ZEB1 slows tumor growth and metastasis (Chen D Int J Gynecol Cancer 2013). However, this is paradoxical as miR-200 family members are expressed at low or negligible levels in normal cells and substantially increase in expression in ovarian cancer, whereas expression of ZEB1 and ZEB2 shows the opposite pattern (Bendoraitis A Gynecol Oncol 2010). How might the authors reconcile these roles for low miR-200 as a marker for both normal tissue and for aggressive cancer but not for the transition phase?

Response: Thanks for pointing this out. The well-described double-negative feedback loop between miR-200 family members and ZEB1 in OC is implicated as molecular switch regulating the fine-tuning and reversibility of EMT (Brabletz S, Brabletz T. EMBO Rep 2010). As suggested by the reviewer, in the revised version we deeply discuss the paradoxical roles of miR200/ZEB circuit in normal and transformed OC cells. As reported by Bendoraitis A et al., normal cells on the ovarian surface are of mesothelial origin, displaying mesenchymal and epithelial characteristics. These ovarian surface mesothelial cells initially undergo a mesothelial-to-epithelial transition (Meso-ET), acquiring characteristics of epithelial cells switching from a miR-200^{LOW} and ZEB1/2^{HIGH} state to a miR-200^{HIGH} and ZEB1/2^{LOW} phenotype. Subsequently, EMT may occur at later stages of ovarian cancer progression. In line with these results, OC cells, displaying aggressive traits, express miR-200^{LOW} and ZEB1^{HIGH}, along with ET_AR^{HIGH} and E-cadherin^{LOW}, consistent with a mesenchymal cell phenotype, predominant in aggressive cancers (Diepenbruck & Christofori, 2016). In this conceptual framework, it is increasingly evident the existence of a network of factors being affected by, and affecting, the critical pathways of the pre-cancer or cancer cells, which change cell fate with major implications of key EMT factor interactions and interconnections in cancer progression. Accordingly, we observed that the mutually exclusive relationship between miR-200b/c and ZEB1 is regulated by ET-1/ET_AR axis in mesenchymal OC cells, in HG-SOC cells, that carry a hot spot missense TP53 mutation, and in platinum resistant OC cells. A full understanding of the regulatory effects of miR-200/ZEB and the processes underlying acquisition of these regulatory controls will be essential to the successful future clinical applications of miR200/ZEB circuit in cancer.

Comment 2: The authors chose to focus on SKOV3 and HEY 8A cells (here denoted simply as HEY). From a molecular biology perspective, this is fine as they express Vimentin and might be thought of as more mesenchymal cell lines. I think from the vantage point of illustrating a signaling pathway, cell line choice is less important. The authors should give the caveat, however, that the fidelity of these lines to high-grade serous ovarian cancer is low, as neither cell line has a tp53 mutation, and they likely represent a non-serous histology. The correlation with TCGA helps a little bit in this regard, but testing their hypotheses in a tp53 mutated line would give a better sense of the translational applications for targeting ZEB1 as a therapeutic approach. Is this response limited to mesenchymal type cells? Either way, the in vivo experiments are limited to a single cell line. Performing a set of in vivo experiments with a tp53 mutated cell line would be informative whether or not a response was seen.

Response: We thank the reviewer for this thoughtful and constructive comment pointing to strengthen the clinical relevance of the blockade of ET-1R signaling and ZEB1 in high-grade serous ovarian cancer (HG-SOC) characterized by TP53 mutation. To test our hypotheses in a TP53 mutated HG-SOC cell line of and to provide a greater value of the translational applications for targeting ZEB1 as a therapeutic approach, we performed a new set of new in vitro and in vivo experiments with a common HG-SOC cell line, OVCAR-3, that carries a hot spot missense TP53 mutation (R248Q). The new results included in the revised version of the manuscript demonstrate that macitentan inhibits the expression of ZEB1 and enhances the miR200-b/-c in these cells (new Suppl. Fig. 6a,b). Notably it reduces the number of tumor nodules in HG-SOC OVCAR-3 xenografts (Fig.7a). This effect was paralleled by a robust increase of E-cadherin protein expression levels, and by reduced levels of ZEB1, as evidenced by immunoblotting analysis of metastatic nodules of OVCAR-3 xenografts (new Fig 7b and Suppl. Fig. 6d). This highlights the therapeutic relevance of blocking ET-1R/miR200/ZEB1 circuit by macitentan in HG-SOC with hot spot missense TP53 mutations. The levels of EMT marker and of miR200b/c and ZEB1 expression in the metastatic nodules of OVCAR-3 xenografts untreated and treated with macitentan have been now included in new fig. 7c and new Fig.7d.

Comment 3: In a similar vein regarding translational utility, ZEB1 has been associated with chemoresistance in ovarian cancer (Sakata J Oncotarget 2017; Cui Y Chemotherapy 2018). What is the potential role of macitentan in chemoresistant ovarian cancer or in reversing chemoresistance?

Response: We appreciate the reviewer for this insightful comment. Improved understanding of how pathways operate and interact in the complex network governing the EMT and chemoresistance of OC cells, may provide new strategies to develop more effective treatments (Diepenbruck & Christofori, 2016). We previously demonstrated the inhibitory effect of macitentan on the tumor growth of chemoresistant ovarian cancer cell lines, as A2780 CIS cells that overexpress ET_AR, re-sensitizing tumor cells to cisplatin (Rosanò L et al. Cancer Res. 2014). To test the translational potential of macitentan in chemoresistant ovarian cancer, we performed a new set of in vitro and in vivo experiments with A2780 cells sensitive and resistant to cisplatin (A2780 CIS). The new results included in the revised version of the manuscript demonstrate that macitentan hampers the ET-1-induced expression of ZEB1 and enhances the levels of miR200-b/-c, even in A2780 CIS cells that, as expected showed higher levels of ZEB1 (new Suppl. Fig. 6a,b). Of note, macitentan is able to reduce the number of tumor nodules in A2780 and A2780 CIS xenografts (Fig.7a). This inhibitory effect was paralleled by decreased ZEB1 protein expression levels, as evidenced by immunoblotting analysis (Fig. 7b and Suppl. Fig.6d), and increased levels of miR-200b/c and EMT markers, as observed in RT-PCR analysis, of metastatic nodules of A2780 and A2780 CIS xenografts (new

Fig 7c,d), highlighting the therapeutic relevance of blocking ET-1R/miR200/ZEB1 circuitry by macitentan in chemoresistant ovarian cancer.

Comment 4: In the discussion, the authors should note that endothelin-1 is a secondary target for bevacizumab. Supporting the authors hypothesis of endothelin-1 as a therapeutic target in mesenchymal subtype ovarian cancer, the mesenchymal subtype derives relatively greater benefit from bevacizumab than immunoreactive or differentiated subtypes

Response: We thank the reviewer for this suggestion that is now added in the Discussion section. The inter-relationship between ET-1 and VEGF-A was deeply studied demonstrating that bevacizumab influences ET-1 plasma level (Kaseb AO et al. Oncology 2012; Dirican A et al. Med Oncol 2014). It has been previously reported that treatment with bevacizumab is of greater benefit for those patients with the mesenchymal subtype of ovarian cancer (Kommos S. et al. Clin Cancer Res. 2017), in which ET_AR and ZEB1 are overexpressed. These results raise the question as to whether the combination of an anti-VEGF therapy with an ET-1R antagonist might improve prognosis of this subgroup of OC patients, representing an important therapeutic options for these patients. We can envisage that bevacizumab may be more effective in combination with ET-1R antagonist in this molecular subtype, targeting the miR200/ZEB1 network which have a critical role in the aggressiveness of OC.

Reviewer #3

Comment 1: Despite authors determined the binding and regulatory potential of miR200b/c on ZEB1, the experiments are especially designed to define such interactions, as reporter assay for UTR sequences. Therefore, it may be discussed whether such interactions are relevant in an in vivo context with endogenous proteins. It should be interesting to evaluate the latter by CHIP assay in primary cultures and/or CO cell lines. In Figure 4 authors show that ZEB1 regulates ETaR through repressing miR200b/c. The experiments are right and clearly show the regulatory feedback for expression of both proteins, ETaR and ZEB1, as well as miR200b/c. However, due to experimental conditions, it is necessary to establish the in vivo significance of the suggested mechanism considering the endogenous components. Authors should also discuss the latter due to the complex nature of signaling pathways and their diversity related to EMT in which an unique miR can have many has potential effects.

Response: As suggested by the reviewer, we performed CHIP assay to evaluate the binding of ZEB1 on the promoter of miR200b. As shown in the new Suppl. Fig.4m, ET-1 promotes the recruitment of ZEB1 on miR-200b promoter, indicating the regulatory interactions promoted by ET-1 on the transcriptional activity of miR-200b. Most relevantly, we addressed whether such interactions are relevant in an in vivo context with endogenous proteins, by analysing this interaction in the nodules tissues of ovarian cancer xenografts treated and untreated with macitentan (new Fig 7e). Notably, CHIP assays on nuclear extracts from metastatic nodules of SKOV3 xenografts, revealed the recruitment of ZEB1 on miR200b promoter and treatment with macitentan impaired this recruitment restraining the regulatory interaction of ZEB1 on the transcriptional activity of miR-200 (new Fig. 7e). These in vivo results, along with new in vivo results obtained in the metastatic nodules of different OC xenografts, strengthen the clinical relevance of the integrated signaling network between ET_AR/ZEB-1 and miR-200b/c in ovarian cancer that can be hampered by macitentan.

As suggested also by the reviewer#2 in comment 1, we highlighted, in the Discussion section, the peculiar complex and interdependent nature of the ZEB1/miR-200 network being integrated with the ET-1/ET_AR pathways, with major implications in the change of OC cell fate.

Comment 2: In figure 6 the role of ETaR-miR200b/c-ZEB1 pathway in ET-1 activation via EMT, cell plasticity and invasion are quite clear. However, it is unclear why authors did not

use ET_AR silencing, as indeed previously used (eg. Figs 2g and 5a). Why did they just perform pharmacological inhibition (macitentan) of ET_AR?

Response: Thanks for pointing this out. We added in the new Fig. 6 c-f and new Suppl. 5b-e, the results of the experiments carried out upon ET_AR silencing. Moreover, as showed in the new Suppl. Fig. 4d, we analyzed the effect of selective ET_AR antagonist zibotentan (ZD4054), and of selective ET_BR antagonist BQ788, on the expression of ZEB1, clearly demonstrating the specific role of ET_AR in the suggested mechanism of OC cells.

Comment 3: Concluding with a glimpse of a therapeutic strategy derived from the findings is quite welcome. However, authors should discuss about possible pharmacological alternatives with higher specificity for ET_AR, especially decreasing a potential side effect of macitentan. In addition, authors should discuss the extent that ET_BR may have a role in the suggested mechanism.

Response: As reported in the new Suppl. Fig. 4d, the selective ET_BR antagonist seems to have only marginal role in the inhibition of ZEB1, suggesting that ET_AR/ET-1 axis represents the main guidance cue in ovarian cancer cells. As suggested by the reviewer, we discussed the possible pharmacological alternatives with higher specificity for ET_AR compared to the dual ET_AR/ET_BR antagonists. Selective ET_AR antagonists have been evaluated for their antitumor activity in human clinical trials, with no measurable, statistically significant advantages, even though the drugs were well-tolerated (Rosanò L. et al. Nature Rev. Cancer 2013; Enevoldsen FC. et al. J Clin Med. 2020; Barton, M. and Yanagisawa, M. Hypertension 2019). In this regard, it is possible that dual antagonists are more appropriate than single antagonists in cancer treatment. Tumor microenvironment (TME) elements, including cancer-associated fibroblasts and tumor-associated macrophages, are mandatory for human tumor progression, and these cells express both ET_AR and ET_BR. These aspects were not considered in the trials. Indeed, the therapeutic benefit of the dual ET-1R antagonist is to target not only OC cells expressing mainly ET_AR, hampering the ZEB1/miR200 network, but also to interfere with TME elements, such as vascular, lymphatic, fibroblasts and inflammatory cells, which mainly express ET_BR, representing therefore an enhanced therapeutic benefit compared to selective ET_AR antagonist, that can impair ZEB1/mir200 transcriptional machinery and the complex regulatory circuits in the stromal architecture. Future preclinical studies should be conducted to design successful trials reflecting the full therapeutic potential of ET-1R antagonists.

REVIEWERS' COMMENTS:

Reviewer #2 (Remarks to the Author):

The role of EMT in ovarian cancer pathogenesis and progression remains a fascinating topic and one in need of continued work. In this manuscript, the authors posit a novel feedback loop between two previously identified signaling pathways, ET-1/ETAR and miR-200/ZEB1. The data delineating the complex double negative feedback loop are well-presented and the experiments flow well in description. I am satisfied that the authors have addressed my key concerns, namely the addition of the OVCAR3 and ChIP experiments adds considerably to the level of evidence displayed in the paper and strengthens the findings. I now think that the work is acceptable for publication.

Reviewer #3 (Remarks to the Author):

Authors have comprehensively responded to all my comments.

Reviewer #2

Comment: The role of EMT in ovarian cancer pathogenesis and progression remains a fascinating topic and one in need of continued work. In this manuscript, the authors posit a novel feedback loop between two previously identified signaling pathways, ET-1/ETAR and miR-200/ZEB1. The data delineating the complex double negative feedback loop are well-presented and the experiments flow well in description. I am satisfied that the authors have addressed my key concerns, namely the addition of the OVCAR3 and CHIP experiments adds considerably to the level of evidence displayed in the paper and strengthens the findings. I now think that the work is acceptable for publication.

Response: Thanks for your great suggestions. It helps us a lot to improve our work.

Reviewer #3

Comment: Authors have comprehensively responded to all my comments.

Response: Thank you for your previous helpful suggestions. It definitely improved our manuscript.